# PCF Learned Sort: a Learning Augmented Sort Algorithm with $\mathcal{O}(n \log \log n)$ Expected Complexity

**Atsuki Sato**                                                                 *a_sato@hal.t.u-tokyo.ac.jp*
*Graduate School of Information Science and Technology*
*The University of Tokyo*

**Yusuke Matsui**                                                              *matsui@hal.t.u-tokyo.ac.jp*
*Graduate School of Information Science and Technology*
*The University of Tokyo*

**Reviewed on OpenReview:** *https: // openreview. net/ forum? id=wVkb8WHbvR*

## Abstract

Sorting is one of the most fundamental algorithms in computer science. Recently, Learned Sorts, which use machine learning to improve sorting speed, have attracted attention. While existing studies show that Learned Sort is empirically faster than classical sorting algorithms, they do not provide theoretical guarantees about its computational complexity. We propose Piecewise Constant Function (PCF) Learned Sort, a theoretically guaranteed Learned Sort algorithm. We prove that the expected complexity of PCF Learned Sort is $\mathcal{O}(n \log \log n)$ under mild assumptions on the data distribution. We also confirm empirically that PCF Learned Sort has a computational complexity of $\mathcal{O}(n \log \log n)$ on both synthetic and real datasets. This is the first study to theoretically support the empirical success of Learned Sort, and provides evidence for why Learned Sort is fast.

## 1 Introduction

Sorting is one of the most fundamental algorithms in computer science and has been extensively studied for many years. Recently, a novel sorting method called Learned Sort has been proposed (Kraska et al., 2019). In Learned Sort, a machine learning model is trained to estimate the distribution of elements in the input array, specifically, the cumulative distribution function (CDF). The model's predictions are then used to rearrange the elements, followed by a minor refinement step to complete the sorting process. Empirical results show that Learned Sort is faster than classical sorting algorithms, including highly optimized counting-based sorting algorithms, comparison sorting algorithms, and hybrid sorting algorithms.

On the other hand, there are few theoretical guarantees regarding the computational complexity of Learned Sort. The first proposed Learned Sort algorithm (Kraska et al., 2019) has a best-case complexity of $\mathcal{O}(n)$, but its expected or worst-case complexity is not discussed. The more efficient Learned Sort algorithms proposed later (Kristo et al., 2020; 2021a) also have $\mathcal{O}(n)$ best-case complexity, but $\mathcal{O}(n^2)$ worst-case complexity (or $\mathcal{O}(n \log n)$ with some modifications). The goal of this paper is to develop a Learned Sort that is theoretically guaranteed to be computationally efficient.

We propose Piecewise Constant Function (PCF) Learned Sort, which can sort with an expected complexity $\mathcal{O}(n \log \log n)$ under mild assumptions on the data distribution. In addition, we show that PCF Learned Sort admits a worst-case complexity guarantee that depends on the choice of the internal sorting algorithm. For instance, if MergeSort—whose worst-case complexity is $\mathcal{O}(n \log n)$—is used, then the worst-case complexity of PCF Learned Sort is $\mathcal{O}(n \log n)$. If we instead use the algorithm by Han (2020), which has a worst-case complexity of $\mathcal{O}(n \sqrt{\log n})$, then the worst-case complexity of PCF Learned Sort becomes $\mathcal{O}(n \sqrt{\log n})$. We then empirically confirm that our Learned Sort can sort with a complexity of $\mathcal{O}(n \log \log n)$ on both synthetic and real datasets.

Independently and concurrently, Zeighami & Shahabi (2024) explored complexity-guaranteed Learned Sort. While our PCF Learned Sort is motivated by similar principles and incorporates comparable design choices, several key distinctions exist between our approach and theirs. A comprehensive comparison between our method and that of Zeighami & Shahabi (2024) is provided in Section 5.

This paper is organized as follows. Section 2 reviews related literature. Section 3 introduces PCF Learned Sort and provides complexity theorems with proof sketches. Section 4 provides empirical validation of the theoretical results. We discuss implications and limitations in Section 5, and conclude in Section 6.

## 2 Related Work

Our research is in the context of algorithms with machine learning (Section 2.1). There are two types of sorting algorithms, comparison sorts (Section 2.2) and non-comparison sorts (Section 2.3), and our proposed method is a non-comparison sort. However, the idea, implementation, and proof of computational complexity of our method are similar to those of sample sort, which is a type of comparison sort. Furthermore, our proposed algorithm and proof of computational complexity are based on those of Learned Index (Section 2.4).

### 2.1 Algorithms with Machine Learning

Our work lies in the emerging area of Learned Sort, where machine learning models accelerate sorting. The first approach by Kraska et al. (2019) trained a model $\tilde{F}(q)$ to approximate the CDF $F(q)$ and placed each element at position $n\tilde{F}(x)$, and then applies insertion sort to complete the ordering. While this yields $\mathcal{O}(n)$ in favorable cases, the insertion-based refinement causes a $\mathcal{O}(n^2)$ worst case. Later implementations improved cache efficiency (Kristo et al., 2020), robustness to duplicates (Kristo et al., 2021a), and even introduced cache-friendly CDF models tailored for sorting (Ferragina & Odorisio, 2025), but still relied on insertion sort and thus retained a quadratic worst-case complexity. Using stronger fallback algorithms such as Introsort (Musser, 1997) or TimSort (McIlroy, 1993) reduces this to $\mathcal{O}(n \log n)$, though this only matches classical comparison-based sorts and does not fully explain the empirical advantage of Learned Sort.

Parallelization has also been explored: Carvalho (2022) proposed IPLS, integrating learned partitioning into IPS4o (Axtmann et al., 2017), and Carvalho & Lawrence (2023) framed LearnedSort as a sample sort with a parallel IPS4o implementation. These studies primarily highlight the engineering potential of learned sorting in parallel settings, which is complementary but orthogonal to our focus on providing theoretical guarantees for the sequential case.

In a broader context, Li et al. (2020) showed that neural programs can achieve strong generalization and replicate efficient algorithmic behaviors such as sorting, demonstrating that machine learning can recover algorithmic complexity classes like $\mathcal{O}(n \log n)$ without explicit manual design. This provides a useful perspective on the generalizability of learned algorithms.

Related to this, the field of *algorithms with predictions* studies how machine learning predictions accelerate classical algorithms (Mitzenmacher & Vassilvitskii, 2022), with applications in caching (Narayanan et al., 2018; Rohatgi, 2020; Lykouris & Vassilvitskii, 2021; Im et al., 2022), ski rental (Purohit et al., 2018; Gollapudi & Panigrahi, 2019; Shin et al., 2023), scheduling (Gollapudi & Panigrahi, 2019; Lattanzi et al., 2020; Lassota et al., 2023), and matching (Antoniadis et al., 2023; Dinitz et al., 2021; Sakaue & Oki, 2022). Sorting with predictions has also been analyzed (Lu et al., 2021; Chan et al., 2023; Erlebach et al., 2023), including tight guarantees by Bai & Coester (2023). In the context of algorithms with predictions, machine learning predictions are typically assumed to be available at no cost, and the models are treated as an opaque box. This exclusion contrasts with our problem setting, where we ensure that the computational complexity covers the entire process from receiving an unsorted array to returning a sorted array.

### 2.2 Comparison Sorts

Sorting algorithms that use comparisons between keys and require no other information about the keys are called comparison sorts. It is well-known that the worst-case complexity of a comparison sort is at least $\Omega(n \log n)$. Commonly used comparison sorting algorithms include QuickSort, heap sort, merge sort,

and insertion sort. The GNU Standard Template Library in C++ uses Introsort (Musser, 1997), an algorithm that combines QuickSort, heap sort, and insertion sort. Java (Java, 2023) and Python up to version 3.10 (Peters, 2002) use TimSort (McIlroy, 1993), an improved version of merge sort. Python 3.11 and later use Powersort (Munro & Wild, 2018), a merge sort that determines a near-optimal merge order.

Sample sort (Frazer & McKellar, 1970) extends QuickSort by using multiple pivots instead of one. Sample sort samples a small number of keys from the array, determines multiple pivots, and uses them to partition the array into multiple buckets. The partitioning is repeated recursively until the array is sufficiently small. Among its implementations, the in-place parallel superscalar sample sort (Axtmann et al., 2017) was introduced as a highly efficient parallel algorithm, and later engineering refinements further improved its performance (Axtmann et al., 2022). Its computational and cache efficiency is theoretically guaranteed by a theorem about the probability that the pivots partition the array (nearly) equally.

### 2.3   Non-Comparison Sorts

Non-comparison sorts use information other than key comparison. Radix sort and counting sort are the most common types of non-comparison sorts. Radix sort uses counting sort as a subroutine for each digit. When the number of digits in the array element is $w$, the computational complexity of radix sort is $\mathcal{O}(wn)$. Thus, radix sort is particularly effective when the number of digits is small. There are several variants of radix sort, such as Spreadsort (Ross, 2002), which integrates the advantages of comparison sort into radix sort and is implemented in the Boost C++ Libraries, and RegionSort (Obeya et al., 2019), which enables efficient parallelization by modeling and resolving dependencies among the element swaps.

In addition, non-comparison sorting algorithms tailored for specific data types have been developed. For integer arrays, a deterministic algorithm with worst-case complexity of $\mathcal{O}(n \log \log n)$ (Han, 2002) and a randomized algorithm with expected complexity of $\mathcal{O}(n\sqrt{\log \log n})$ (Han & Thorup, 2002) have been proposed. For real-valued arrays, recent advances have led to the development of a sorting algorithm with a worst-case complexity of $\mathcal{O}(n\sqrt{\log n})$ (Han, 2020). Our PCF Learned Sort also targets real-valued arrays and, under mild assumptions on the distribution, achieves an expected complexity of $\mathcal{O}(n \log \log n)$. Moreover, when the algorithm of Han (2020) is used as the internal sorting component, PCF Learned Sort has a worst-case complexity of $\mathcal{O}(n\sqrt{\log n})$, which matches that of Han (2020).

### 2.4   Learned Index

Kraska et al. (2018) showed that index data structures such as B-trees and Bloom filters can be made faster or more memory efficient by combining them with machine learning models and named such novel data structures Learned Index. Since then, learning augmented B-trees (Wang et al., 2020; Kipf et al., 2020; Ferragina & Vinciguerra, 2020; Zeighami & Shahabi, 2023), Bloom filters (Mitzenmacher, 2018; Vaidya et al., 2021; Sato & Matsui, 2023; 2024), and even range-minimum query structures (Ferragina et al., 2025) have been proposed. There are several works on learning augmented B-trees whose performance is theoretically guaranteed. PGM-index (Ferragina & Vinciguerra, 2020) is a learning augmented B-tree that is guaranteed to have the same worst-case query complexity as the classical B-tree, i.e., $\mathcal{O}(\log n)$. Zeighami & Shahabi (2023) proposed a learning augmented B-tree with an expected query complexity of $\mathcal{O}(\log \log n)$ under mild assumptions on the distribution. More broadly, this line of research connects to earlier studies on compressed data structures with provable guarantees, such as the foundational work of Sadakane (2007).

## 3   Methods

This section introduces our *Learned Sort framework* and its PCF-based instantiation. In Section 3.1, we set up the notation, in Section 3.2 we present the general framework, and in Section 3.3 we instantiate it with a PCF-based CDF model to obtain concrete complexity guarantees.

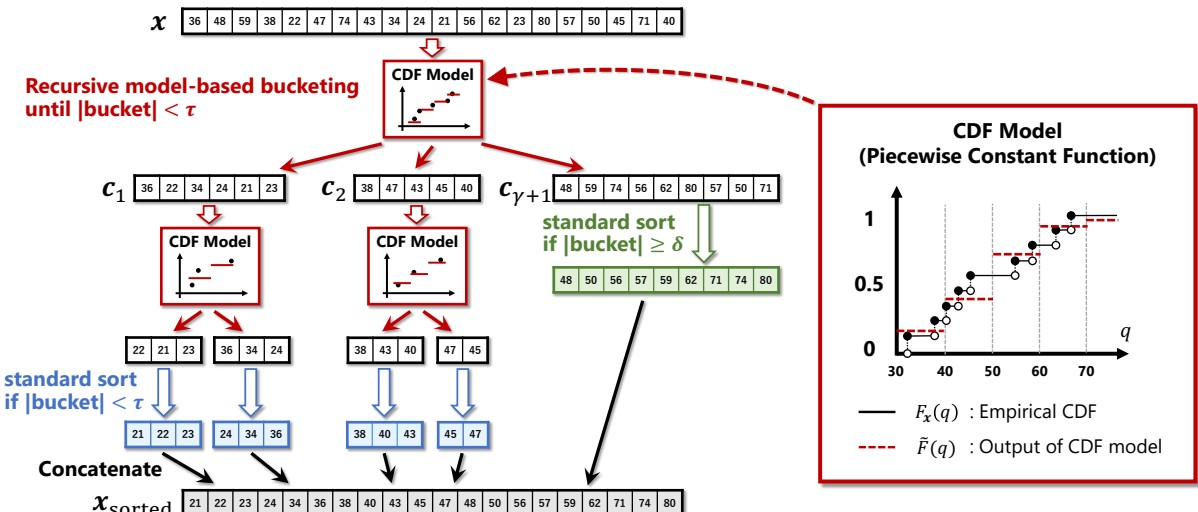

Figure 1: PCF Learned Sort: First, the input array is partitioned into $\gamma + 1$ buckets using a CDF model-based method. Buckets larger than $\delta$ or smaller than $\tau$ are sorted with a standard sort (e.g., IntroSort or QuickSort). Otherwise, the recursive model-based bucketing is repeated. Finally, the sorted arrays are concatenated. The CDF model used for bucketing is a Piecewise Constant Function (PCF). The function is constant within each interval, and the interval widths are constant.

## 3.1 Notation and Setup

Following the learned-sort paradigm Kraska et al. (2019); Kristo et al. (2020; 2021a), our learned sort algorithm recursively applies *model-based bucketing*. Model-based bucketing partitions the array into multiple buckets using a CDF model trained on the input so that the bucket with the larger ID gets the larger value.

Let $\mathcal{D} \subseteq \mathbb{R}$ be the domain of possible key values and $\boldsymbol{x} \in \mathcal{D}^n$ be the input array. For vector/array $\boldsymbol{v}$, we write $|\boldsymbol{v}|$ for its length (number of elements), e.g., $|\boldsymbol{x}| = n$. If $|\boldsymbol{x}|$ is smaller than a fixed threshold $\tau \in \mathbb{N}$, the algorithm falls back to a classic sorting algorithm. We refer to this fallback classical sorting algorithm as the *standard sort*. As the standard sort, we can use any sorting algorithm, such as IntroSort, QuickSort, or the algorithm of Han (2020).

For model-based bucketing, we define functions $\alpha(n), \beta(n), \gamma(n), \delta(n) : \mathbb{N} \to \mathbb{N}$ as follows: $\alpha(n)$ is the number of samples used to train the CDF model, $\beta(n)$ is the number of PCF bins (a hyperparameter; see Section 3.3), $\gamma(n) + 1$ is the number of buckets, and $\delta(n)$ is the threshold that determines the behavior after bucketing (see Section 3.2). For brevity, we write $\alpha, \beta, \gamma, \delta$.

## 3.2 Learned-Sort Framework

**Algorithm description.** At a high level, our algorithm is a recursive model-based bucketing scheme augmented with an exception-handling mechanism for bucketing "failures." This exception-handling mechanism greatly simplifies the analysis of both expected and worst-case complexity. The overall workflow is visualized in Figure 1, and the pseudocode is given in Algorithm 1.

If the length of the input array is less than $\tau$, our algorithm sorts the input array using a standard sort (e.g., IntroSort or QuickSort). Otherwise, model-based bucketing is performed.

The model-based bucketing method $\mathcal{M}$ takes an input array $\boldsymbol{x}$ and partitions it into several buckets. First, all or some elements of $\boldsymbol{x}$ are used to train the CDF model $\tilde{F} : \mathcal{D} \to [0, 1]$. The $\tilde{F}(q)$ is trained to approximate the empirical CDF $F_{\boldsymbol{x}}(q) = |i \in \{1, \ldots, n\} \mid x_i \leq q|/|\boldsymbol{x}|$. Any non-decreasing model can serve as the CDF model $\tilde{F}(q)$ (e.g., linear models, monotonic MLPs, or the PCF introduced in Section 3.3).

---

**Algorithm 1** The Learned Sort algorithm

---

1: **Input:** $\boldsymbol{x} \in \mathbb{R}^n$     **Output:** $\boldsymbol{x}_{\text{sorted}} \in \mathbb{R}^n$ (the sorted version of $\boldsymbol{x}$)

2: **Subroutines:**

3:     STANDARD-SORT($\boldsymbol{x}$) : The standard sort with complexity guarantees (e.g, IntroSort or QuickSort)

4:     CDF-MODEL($\boldsymbol{x}$) : Instantiate a CDF Model $\tilde{F}(q)$ that estimates $F_{\boldsymbol{x}}(q)$

5: **function** LEARNED-SORT($\boldsymbol{x}$)

6:     $n \leftarrow |\boldsymbol{x}|$

7:     **if** $n < \tau$ **then**                                         ▷ Small bucket

8:        **return** STANDARD-SORT($\boldsymbol{x}$)

9:     $\tilde{F}(q) \leftarrow$ CDF-MODEL($\boldsymbol{x}$)                           ▷ Model-based bucketing

10:     $\boldsymbol{c}_1 \leftarrow [\,], \; \boldsymbol{c}_2 \leftarrow [\,], \; \ldots \;, \; \boldsymbol{c}_{\gamma+1} \leftarrow [\,]$

11:     **for** $i = 1, 2, \ldots, n$

12:        $j \leftarrow \lfloor \tilde{F}(x_i)\gamma \rfloor + 1$

13:        $\boldsymbol{c}_j$.APPEND($x_i$)

14:     **for** $j = 1, 2, \ldots, \gamma+1$                  ▷ Recursively sort and concatenate

15:        **if** $|\boldsymbol{c}_j| \geq \delta$ **then**

16:           $\boldsymbol{c}_j \leftarrow$ STANDARD-SORT($\boldsymbol{c}_j$)

17:        **else**

18:           $\boldsymbol{c}_j \leftarrow$ LEARNED-SORT($\boldsymbol{c}_j$)

19:     $\boldsymbol{x}_{\text{sorted}} \leftarrow$ CONCATENATE($\boldsymbol{c}_1, \boldsymbol{c}_2, \ldots, \boldsymbol{c}_{\gamma+1}$)

20:     **return** $\boldsymbol{x}_{\text{sorted}}$

---

After training, the CDF model is used to partition the input array $\boldsymbol{x}$ into $\gamma + 1$ buckets. All $\gamma + 1$ buckets, $\{\boldsymbol{c}_j\}_{j=1}^{\gamma+1}$, are initialized to be empty, and then for each $i \in \{1, \ldots, n\}$, $x_i$ is appended to $\boldsymbol{c}_{\lfloor \tilde{F}(x_j)\gamma \rfloor + 1}$. This is based on the intuition that the number of elements less than or equal to $x_i$ in the array $\boldsymbol{x}$ (i.e., $|\{j \in \{1, \ldots, n\} \mid x_j \leq x_i\}|$) is approximately equal to $n\tilde{F}(x_i)$.

We restrict the CDF model $\tilde{F}$ to non-decreasing functions to ensure that the bucket with the larger ID gets the larger value, i.e., $p \in \boldsymbol{c}_j \; \wedge \; q \in \boldsymbol{c}_k \; \wedge \; j < k \;\; \Rightarrow \;\; p < q$. This means that each bucket is responsible for a disjoint and continuous interval. Let $t_j = \min_{x \in \boldsymbol{c}_j} x$ ($j = 1, \ldots, \gamma+1$), $t_{\gamma+2} = \infty$, then the $j$-th bucket $\boldsymbol{c}_j$ ($j = 1, \ldots, \gamma+1$) is responsible for a continuous interval $\mathcal{I}_j \coloneqq [t_j, t_{j+1})$.

After model-based bucketing, our algorithm determines for each bucket whether the bucketing "succeeds" or "fails." For each $j \in \{1, \ldots, \gamma+1\}$, we check whether the size of bucket $\boldsymbol{c}_j$ is less than $\delta$. If $|\boldsymbol{c}_j| \geq \delta$ (which means the bucketing "fails"), the bucket is sorted using the standard sort (e.g., IntroSort or QuickSort). If $|\boldsymbol{c}_j| < \delta$ (which means the bucketing "succeeds"), the bucket is sorted by recursively calling our Learned Sort algorithm. Note that the parameters such as $\gamma$ and $\delta$ are redetermined for each recursion according to the size of the bucket (i.e., the input array in the next recursion step), and the CDF model is retrained for each bucket. Finally, the sorted buckets are concatenated to form the sorted array.

**Worst-case complexity.** The following lemma guarantees the worst-case complexity of our Learned Sort.

**Lemma 3.1.** *Assume that there exists a model-based bucketing algorithm $\mathcal{M}$ such that $\mathcal{M}$ can perform bucketing (including model training and inferences) an array of length $n$ into $\gamma + 1 = \mathcal{O}(n)$ buckets with a worst-case complexity of $\mathcal{O}(n)$. Also, assume that the standard sort has a worst-case complexity of $\mathcal{O}(nU(n))$, where $U(n)$ is a non-decreasing function. Then, the worst-case complexity of our Learned Sort with such $\mathcal{M}$ and $\delta = \lfloor n^d \rfloor$ (where $d$ is a constant satisfying $0 < d < 1$) is $\mathcal{O}(nU(n) + n \log \log n)$.*

This lemma can be intuitively shown from the following two points: (1) the maximum recursion depth is $\mathcal{O}(\log \log n)$, and (2) each element of the input array $\boldsymbol{x}$ undergoes several bucketing and only one standard sort. The first point can be shown from the fact that the size of the bucket in the $i$-th recursion depth is less than $n^{d^i}$. The second point is evident from the algorithm's design since the buckets sorted by standard sort are now left only to be concatenated. The exact proof is given in Appendix A.1.

Note that this guarantee critically relies on the exception-handling mechanism: without it, the recursion depth could reach $\Theta(n)$ and the worst-case time would degrade to $\Theta(n^2)$, whereas with it the recursion depth is always bounded by $O(\log \log n)$.

**Expected complexity.** Next, we introduce a lemma about the expected computational complexity of our Learned Sort. The following assumption is necessary to guarantee the expected computational complexity.

**Assumption 3.2.** The input array $\boldsymbol{x} \in \mathcal{D}^n$ is formed by independent sampling according to a probability density distribution $f(x) \colon \mathcal{D} \to \mathbb{R}_{\geq 0}$.

We define $f_{\mathcal{I}}(x) \colon \mathcal{I} \to \mathbb{R}_{\geq 0}$ to be the conditional probability density distribution of $f(x)$ under the condition that $x \in \mathcal{I}$ for a interval $\mathcal{I} \subseteq \mathcal{D}$, i.e., $f_{\mathcal{I}}(x) \coloneqq \frac{f(x)}{\int_{\mathcal{I}} f(y)dy}$.

The expected computational complexity of our proposed Learned Sort is guaranteed by the following lemma.

**Lemma 3.3.** *Let $\boldsymbol{x}_{\mathcal{I}} \ (\in \mathcal{I}^n)$ be the array formed by sampling $n$ times independently according to $f_{\mathcal{I}}(x)$. Assume that there exist a model-based bucketing algorithm $\mathcal{M}$ and a constant $d \ (\in (0,1))$ that satisfy the following three conditions for any interval $\mathcal{I} \ (\subseteq \mathcal{D})$: (i) $\mathcal{M}$ can perform bucketing (including model training and inferences) on an array of length $n$, with an expected complexity of $\mathcal{O}(n)$, (ii) $\gamma + 1 = \mathcal{O}(n)$, and (iii) $\Pr[\exists j, |\boldsymbol{c}_j| \geq \lfloor n^d \rfloor] = \mathcal{O}\left(\frac{1}{\log n}\right)$. Also, assume that the standard sort has an expected complexity of $\mathcal{O}(n \log n)$. Then, the expected complexity of our Learned Sort with such $\mathcal{M}$ and $\delta = \lfloor n^d \rfloor$ is $\mathcal{O}(n \log \log n)$.*

This lemma can be proved intuitively by the following two points: (1) the maximum recursion depth is $\mathcal{O}(\log \log n)$, and (2) the expected total computational complexity from the $i$-th to the $(i+1)$-th recursion depth is $\mathcal{O}(n)$. The first point is the same as in the explanation of the proof of Lemma 3.1. The second point holds because the expected computational complexity from the $i$-th to the $(i+1)$-th recursion depth is $\mathcal{O}(n \log n)$ with probability $\mathcal{O}(\frac{1}{\log n})$, and $\mathcal{O}(n)$ in other cases. See Appendix A.2 for the exact proof.

Note that the assumption of Lemma 3.3 includes "$\mathcal{M}$ works well with high probability for any $\mathcal{I} \ (\subseteq \mathcal{D})$." This is because our Learned Sort algorithm recursively repeats the model-based bucketing. The range of elements in the bucket, i.e., the input array in the next recursion step, can be any interval $\mathcal{I} \ (\subseteq \mathcal{D})$.

### 3.3 PCF Learned Sort

We now instantiate the framework with *PCF Learned Sort*, which satisfies the assumptions of Lemma 3.1 and Lemma 3.3, thereby providing both worst-case and expected-time guarantees. PCF Learned Sort approximates the CDF using a *Piecewise Constant Function (PCF)* with $\beta$ equal-width bins; the model output is constant within each bin (see the right panel of Figure 1). The study that develops a Learned Index with a theoretical guarantee on its complexity (Zeighami & Shahabi, 2023) also used PCF as a CDF model. While our framework admits more expressive CDF models (such as spline-based or neural models), we focus on PCF due to its minimal training and inference cost and the tractability of its theoretical analysis.

The model-based bucketing method in PCF Learned Sort $\mathcal{M}_{\mathrm{PCF}}$ trains the CDF model $\tilde{F}$ as follows. The parameters $\alpha \in \{1, \dots, n\}$ and $\beta \in \mathbb{N}$ are determined by $n$, where $\alpha$ is the number of samples used to train the model and $\beta$ is the number of intervals in the PCF. The PCF is trained by counting the number of samples in each interval. We define $i(x) = \lfloor (x - x_{\min})\beta/(x_{\max} - x_{\min}) \rfloor + 1$, where $x_{\min} = \min_i x_i$ and $x_{\max} = \max_i x_i$. From $\boldsymbol{x}$, $\alpha$ samples are taken to form $\boldsymbol{a} \in \mathcal{D}^{\alpha}$, and $i(x)$ is used to construct $\boldsymbol{b} \in \mathbb{Z}_{\geq 0}^{\beta+1}$ with $b_i = |\{j \in \{1, \dots, \alpha\} \mid i(a_j) \leq i\}|$. This counting procedure trains the PCF. Note that $\boldsymbol{b}$ is an non-decreasing non-negative array and $b_{\beta+1} = \alpha$, i.e., $0 \leq b_1 \leq b_2 \leq \cdots \leq b_{\beta+1} = \alpha$.

Inference for $\tilde{F}(x)$ is then given by $\tilde{F}(x) = \frac{b_{i(x)}}{\alpha}$. Since $i(x)$ and $\boldsymbol{b}$ are non-decreasing, $\tilde{F}(x)$ is also non-decreasing. Also, $0 \leq \tilde{F}(x) \leq 1$ because $0 \leq b_i \leq \alpha$ for every $i$.

The following is a lemma to bound the probability that $\mathcal{M}_{\mathrm{PCF}}$ will "fail" bucketing. This lemma is important to guarantee the expected computational complexity of PCF Learned Sort.

**Lemma 3.4.** *Let $\sigma_1$ and $\sigma_2$ be respectively the lower and upper bounds of the probability density distribution $f(x)$ in $\mathcal{D}$, and assume that $0 < \sigma_1 \leq \sigma_2 < \infty$. That is, $x \in \mathcal{D} \ \Rightarrow \ \sigma_1 \leq f(x) \leq \sigma_2$. Then, in model-based*

*bucketing of $\boldsymbol{x}_{\mathcal{I}}$ ($\in \mathcal{I}^n$) to $\{\boldsymbol{c}_j\}_{j=1}^{\gamma+1}$ using $\mathcal{M}_{\mathrm{PCF}}$, the following holds for any interval $\mathcal{I}$ ($\subseteq \mathcal{D}$):*

$$K := \frac{\gamma\delta}{2n} - \frac{2\sigma_2\gamma}{\sigma_1\beta} \geq 1 \quad \Rightarrow \quad \Pr[\exists j, |\boldsymbol{c}_j| > \delta] \leq \frac{2n}{\delta}\exp\left\{-\frac{\alpha K}{2\gamma}\left(1 - \frac{1}{K}\right)^2\right\}. \tag{1}$$

The proof of this lemma is based on and combines proofs from two existing studies. The first is Lemma 5.2. from a study of IPS$^4$o (Axtmann et al., 2022), an efficient sample sort. This lemma guarantees the probability of a "successful recursion step" when selecting pivots from samples and using them to perform a partition. This lemma is for the method that does not use the CDF model, so the proof cannot be applied directly to our case. Another proof we refer to is the proof of Lemma 4.5. from a study that addressed the computational complexity guarantee of the Learned Index (Zeighami & Shahabi, 2023). This lemma provides a probabilistic guarantee for the error between the output of the PCF and the empirical CDF. Some modifications are required to adapt it to the context of sorting and to attribute it to the probability of bucketing failure, i.e., $\Pr[\exists j, |\boldsymbol{c}_j| > \delta]$. By appropriately combining the proofs of these two lemmas, Lemma 3.4 is proved. The exact proof is given in Appendix A.3.

Here, we emphasize that the assumption of this lemma, $0 < \sigma_1 \leq \sigma_2 < \infty$, is sufficiently reasonable and "mild" as described in (Zeighami & Shahabi, 2023). It asserts that the probability density function $f(x)$ is both bounded and nonzero over its domain $\mathcal{D}$. This class of distributions covers the majority of real-world scenarios because real-world data is commonly derived from bounded and continuous phenomena, e.g., age, grades, and data over a period of time. Empirically, Zeighami & Shahabi (2023) further suggest that the ratio $\sigma_2/\sigma_1$ tends to remain close to 1 in a wide variety of practical datasets, and even for more challenging cases like OSM, the ratio still appears to remain at most around 20.

Using Lemma 3.1, Lemma 3.3, and Lemma 3.4, we can prove the following theorems.

**Theorem 3.5.** *If $\mathcal{M}_{\mathrm{PCF}}$ is the bucketing method, the worst-case complexity of standard sort is $\mathcal{O}(nU(n))$ (where $U(n)$ is a non-decreasing function), and $\alpha = \beta = \gamma = \delta = \lfloor n^{3/4} \rfloor$, then the worst-case complexity of PCF Learned Sort is $\mathcal{O}(nU(n) + n\log\log n)$.*

*Proof.* When $\alpha = \beta = \gamma = \lfloor n^{3/4} \rfloor$, the computational complexity for model-based bucketing is $\mathcal{O}(n)$ because (i) the PCF is trained in $\mathcal{O}(\alpha + \beta) = \mathcal{O}(n^{3/4})$, and (ii) the total complexity of inference for $n$ elements is $\mathcal{O}(n)$, since the inference is performed in $\mathcal{O}(1)$ per element. Therefore, since $\gamma + 1 = \mathcal{O}(n)$, the worst-case complexity of standard sort is $\mathcal{O}(nU(n))$, and $\delta = \lfloor n^{3/4} \rfloor$, we can prove the worst-case complexity of PCF Learned Sort is $\mathcal{O}(nU(n) + n\log\log n)$ by Lemma 3.1. $\square$

**Theorem 3.6.** *Let $\sigma_1$ and $\sigma_2$ be the lower and upper bounds, respectively, of the probability density distribution $f(x)$ in $\mathcal{D}$, and assume that $0 < \sigma_1 \leq \sigma_2 < \infty$. Then, if $\mathcal{M}_{\mathrm{PCF}}$ is the bucketing method, the expected complexity of standard sort is $\mathcal{O}(n\log n)$, and $\alpha = \beta = \gamma = \delta = \lfloor n^{3/4} \rfloor$, then the expected complexity of PCF Learned Sort is $\mathcal{O}(n\log\log n)$.*

*Proof.* When $\alpha = \beta = \gamma = \lfloor n^{3/4} \rfloor$, the computational complexity for model-based bucketing is $\mathcal{O}(n)$. Since $K = \Omega(\sqrt{n})$ when $\alpha = \beta = \gamma = \delta = \lfloor n^{3/4} \rfloor$, $K \geq 1$ for sufficiently large $n$, and

$$\frac{2n}{\delta}\exp\left\{-\frac{\alpha K}{2\gamma}\left(1 - \frac{1}{K}\right)^2\right\} = \mathcal{O}(n^{\frac{1}{4}}\exp(-\sqrt{n})) \leq \mathcal{O}\left(\frac{1}{\log n}\right). \tag{2}$$

Given that $\gamma + 1 = \mathcal{O}(n)$, the expected complexity of standard sort is $\mathcal{O}(n\log n)$, and $\delta = \lfloor n^{3/4} \rfloor$, it follows from Lemma 3.3 and Lemma 3.4 that the expected complexity of PCF Learned Sort is $\mathcal{O}(n\log\log n)$. $\square$

Note that the exact value of $\sigma_1$ and $\sigma_2$ is not required to run PCF Learned Sort since the parameters for this algorithm, i.e., $\alpha$, $\beta$, $\gamma$, and $\delta$, are determined without any prior knowledge. In other words, PCF Learned Sort can sort in expected $\mathcal{O}(n\log\log n)$ complexity as long as $0 < \sigma_1 \leq f(x) \leq \sigma_2 < \infty$, even if it does not know the exact value of $\sigma_1$ and $\sigma_2$. If $\sigma_1$ and $\sigma_2$ do not satisfy the assumption of Theorem 3.6, the expected complexity of PCF Learned Sort increases to $\mathcal{O}(nU(n) + n\log\log n)$. Such scenarios arise, for

instance, in heavy-tailed distributions, datasets with extremely sparse regions ($\sigma_1 = 0$), or cases where the density is highly concentrated at particular values ($\sigma_2 = \infty$). However, thanks to the worst-case bound in Theorem 3.5, the complexity never exceeds $\mathcal{O}(nU(n) + n \log \log n)$. Thus, the algorithm remains robust even under distributions that do not fully satisfy the assumption.

**Alternative CDF Models.** Our framework is not limited to PCF; it can also accommodate more expressive CDF models. For instance, we can replace PCF with a spline-based model that approximates the CDF by interpolating the empirical distribution at bin boundaries. Concretely, given $\beta$ intervals, we evaluate the empirical CDF at the endpoints of each interval and then construct a piecewise linear spline that connects these values, yielding a continuous and non-decreasing CDF approximation. We refer to the algorithm that applies this spline-based CDF model within our Learned Sort framework as *Spline Learned Sort*. We show that spline-based models constructed in this way still satisfy the assumptions of Lemma 3.3, and thus the expected time complexity guarantee of $\mathcal{O}(n \log \log n)$ remains valid. A detailed theoretical proof for the spline-based case is provided in Appendix A.5.

## 4  Experiments

In this section, we empirically validate our theoretical results. First, in Section 4.1, we confirm that the complexity of PCF Learned Sort is $\mathcal{O}(n \log \log n)$ for both synthetic and real datasets. Then, in Section 4.2, we conduct experiments with various parameter settings and empirically confirm Lemma 3.4, a lemma that bounds the probability of bucketing failure and plays a crucial role in guaranteeing the expected complexity of PCF Learned Sort. Finally, in Section 4.3, we present from sorting time measurements.

**Setup.** We experimented with synthetic datasets created from the following four distributions: uniform ($\mathsf{min} = 0, \mathsf{max} = 1$), normal ($\mu = 0, \sigma = 1$), exponential ($\lambda = 1$), lognormal ($\mu = 0, \sigma = 1$). The input array was generated by independently taking $n$ samples from each distribution. Only the uniform distribution satisfies the theoretical assumptions required for our complexity guarantees. The other distributions violate these assumptions, but we include them to evaluate the empirical robustness of PCF Learned Sort.

We also used the following 16 real datasets, including **Chicago [Start, Tot]** (Chicago, 2021), **NYC [Pickup, Dist, Tot]** (nyc, 2020), **SOF [Humidity, Pressure, Temperature]** (Mavrodiev, 2019), **Wiki**, **OSM**, **Books**, **Face** (Marcus et al., 2020), and **Stocks [Volume, Open, Date, Low]** (Onyshchak, 2020). Further dataset details are provided in Appendix B. For each dataset, we randomly sample $n$ elements, shuffle them, and use them as an input array to examine the complexity of the sort algorithms. Since these are real-world datasets, we cannot definitively determine whether they satisfy our theoretical assumptions. However, as shown in the histograms in Figure 2, Chicago [Start], NYC [Pickup, Tot], SOF [Humidity], and Face appear to distribute values across a relatively dense and continuous domain, aligning well with our assumptions. In contrast, other datasets exhibit long tails or sparse regions, suggesting that our assumptions may not hold. Furthermore, note that for several datasets (Chicago [Start, Tot], NYC [Dist, Tot], SOF [Humidity, Pressure, Temperature], and Stocks [Volume, Open, Date, Low]), the number of unique values is extremely small (less than 3.2% of the total elements, as detailed in Appendix B).

All experiments were run on a Linux machine equipped with an Intel® Core™ i9-11900H CPU @ 2.50GHz and 62GB of memory. GCC version 9.4.0 was used for compilation, employing the `-O3` optimization flag.

### 4.1  Computational Complexity of PCF Learned Sort

We meticulously counted the total number of basic operations for sorting the input array to observe the computational complexity of each sorting algorithm. Here, the basic operations consist of four arithmetic operations, powers, comparisons, logical operations, assignments, and memory access. We chose this metric, which counts basic operations, to mitigate the environmental dependencies observed in other metrics, such as CPU instructions and CPU time, which are heavily influenced by compiler optimizations and the underlying hardware. This is the same idea as the metric selection in the experiment of (Zeighami & Shahabi, 2023).

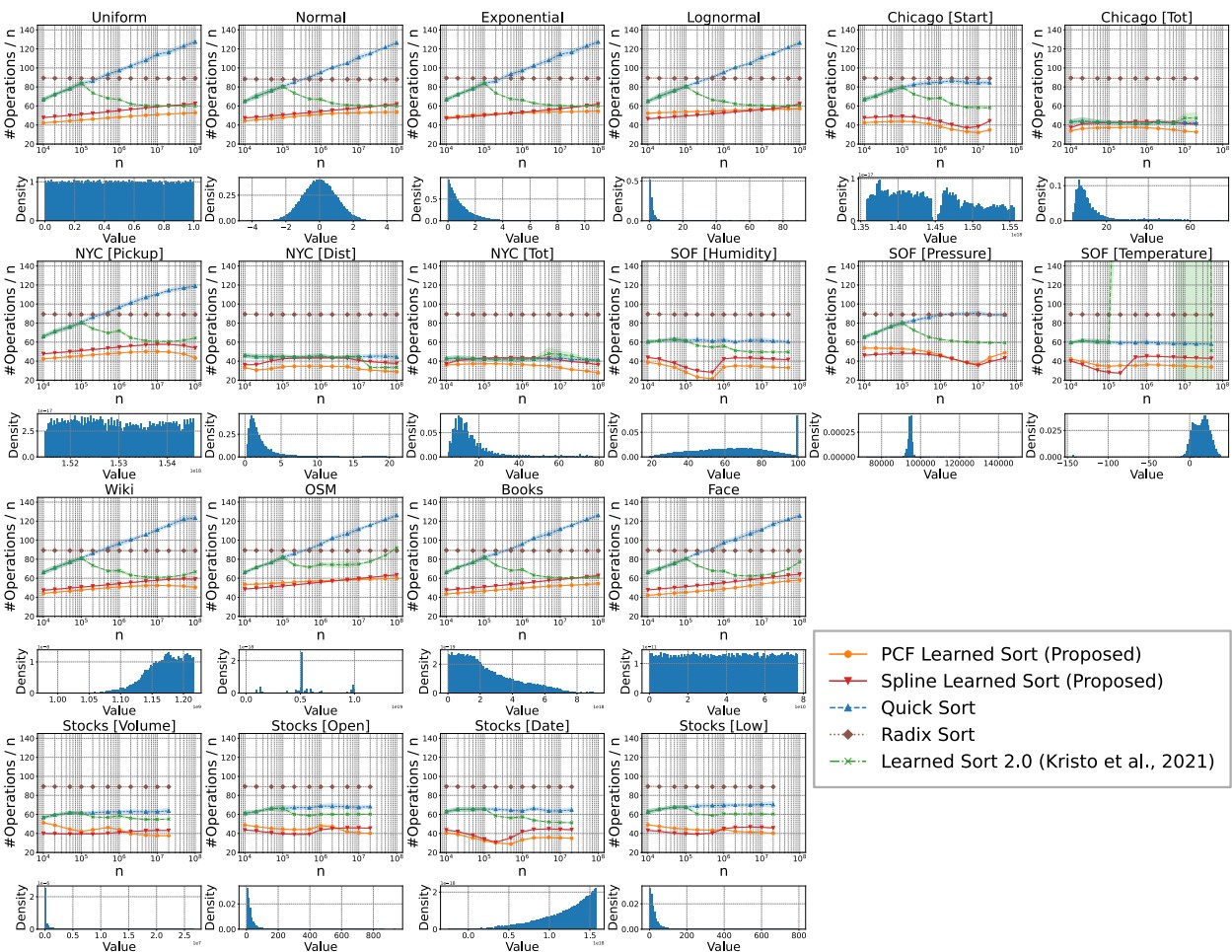

Figure 2: Number of operations to sort the array. Below each graph is a histogram visualizing the distribution of each dataset. The standard deviation of the 10 measurements is represented by the shaded area. Our PCF Learned Sort consistently achieves a complexity lower than $\mathcal{O}(n \log n)$, while Learned Sort 2.0 (Kristo et al., 2021a), which has $\mathcal{O}(n^2)$ worst-case complexity, occasionally requires huge operations.

The parameters of PCF Learned Sort are set as in Theorems 3.5 and 3.6, $\alpha = \beta = \gamma = \delta = \lfloor n^{3/4} \rfloor$ and $\tau = 100$. As the standard sort, we used QuickSort, which has an expected complexity of $\mathcal{O}(n \log n)$. For ease of implementation and straightforward measurement of comparison counts, we chose QuickSort. We compared our PCF Learned Sort against (plain) QuickSort, radix sort, and Learned Sort 2.0 (Kristo et al., 2021b). Learned Sort 2.0 was selected as the learned-method baseline because it allows relatively simple counting of operations (a broader comparison appears in the sorting time experiments of Section 4.3).

Figure 2 shows the number of operations divided by the length of the input array, $n$. Each point represents the average over 10 runs, with the shaded region indicating the standard deviation. Note that the horizontal axis is logarithmic. As a result, the curve for QuickSort (with $\mathcal{O}(n \log n)$ complexity) appears approximately linear in this plot for synthetic datasets and real-world datasets with few duplicates (i.e., NYC [Pickup], Wiki, OSM, Books, and Face). In contrast, the curve for PCF Learned Sort is nearly flat, suggesting a complexity significantly lower than $\mathcal{O}(n \log n)$. PCF Learned Sort consistently requires the fewest operations across all conditions, achieving up to 2.8 times fewer operations than QuickSort. These results confirm not only our theoretical analysis but also the robustness of PCF Learned Sort under assumption-violating scenarios.

Figure 3: Heatmap showing the empirical frequency of bucketing failure, i.e., $\exists j, |\boldsymbol{c}_j| > \delta$. The variables $a, b, c, d$, except those on the x- and y-axes, were set to 0.75. The white dotted line represents the parameters that make the right side of Equation (1) equal to 0.5. The close alignment between this white dotted line and the actual success/fail boundery suggests the theoretical bound by is Equation (1) reasonably tight.

For datasets with many duplicates, QuickSort, Learned Sort 2.0 (Kristo et al., 2021a), and PCF Learned Sort all exhibit relatively few operations. This behavior arises because highly duplicated datasets often result in buckets containing only one distinct value, allowing the algorithms to terminate early. Even under such conditions, PCF Learned Sort performs fewer operations than the other algorithms.

The curve for radix sort is also flat but lies consistently above that of PCF Learned Sort. This is due to the difference in partitioning strategies. While the radix sort partitions at a fixed granularity, our PCF Learned Sort performs partitioning using intervals that are adaptively set by the learning model.

In particular, on the SOF [Temperature] dataset with medium input sizes ($2 \times 10^5 < n \le 2 \times 10^7$), Learned Sort 2.0 incurs extremely high costs: while all other methods keep the number of operations per $n$ below 100, it can exceed 50,000. This shows that Learned Sort methods without worst-case guarantees may suffer from pathological overhead. By contrast, our method maintains both expected and worst-case guarantees, ensuring robust performance across diverse datasets.

## 4.2 Confirmation of Lemma 3.4

Lemma 3.4 bounds the probability that a bucket of size greater than $\delta$ exists. This is an important lemma that allows us to guarantee the expected computational complexity of PCF Learned Sort. Here, we empirically confirm that this upper bound is appropriate.

We have experimented with $\alpha = \lfloor n^a \rfloor, \beta = \lfloor n^b \rfloor, \gamma = \lfloor n^c \rfloor, \delta = \lfloor n^d \rfloor$, varying $a, b, c, d$ from 0.05 to 0.95 at 0.05 intervals. For each $a, b, c, d$ setting, the following was repeated 100 times: we took $n = 10^6$ elements from the uniform distribution to form the input array and divided the array into $\gamma + 1$ buckets by $\mathcal{M}_{\mathrm{PCF}}$, and checked whether or not $\exists j, |\boldsymbol{c}_j| > \delta$. Thus, for each $a, b, c, d \in \{0.05, 0.10, \ldots, 0.95\}$, we obtained the empirical frequency at which bucketing "fails."

Heat maps in Figure 3 show the empirical frequency of bucketing failures when two of the $a, b, c, d$ parameters are fixed, and the other two parameters are varied. The values of the two fixed variables are set to 0.75, e.g., in the upper left heap map of Figure 3 (horizontal axis is $a$ and vertical axis is $b$), $c = d = 0.75$. The white dotted line represents the parameter so that the right side of Equation (1) is 0.5. That is, Lemma 3.4 asserts that "in the region upper right of the white dotted line, the probability of bucketing failure is less than 0.5."

We observe that the white dotted line is close to (or slightly to the upper right of) the actual bound of whether bucketing "succeeds" or "fails" more often. This suggests that the theoretical upper bound from Lemma 3.4 agrees well (to some extent) with the actual probability. We can also confirm that, as Lemma 3.4 claims, the probability of bucketing failure is indeed small in the region upper right of the white line.

## 4.3 Experiments on Sorting Time

We empirically compare the sorting time of our PCF Learned Sort against several baselines. Note that the metric used in Section 4.1, the number of operations, does not change depending on the machine or compilation method, but the sorting time does. As the standard sort used in PCF Learned Sort, we used `std::sort`, which has a worst-case complexity of $\mathcal{O}(n \log n)$. We compared our PCF Learned Sort with

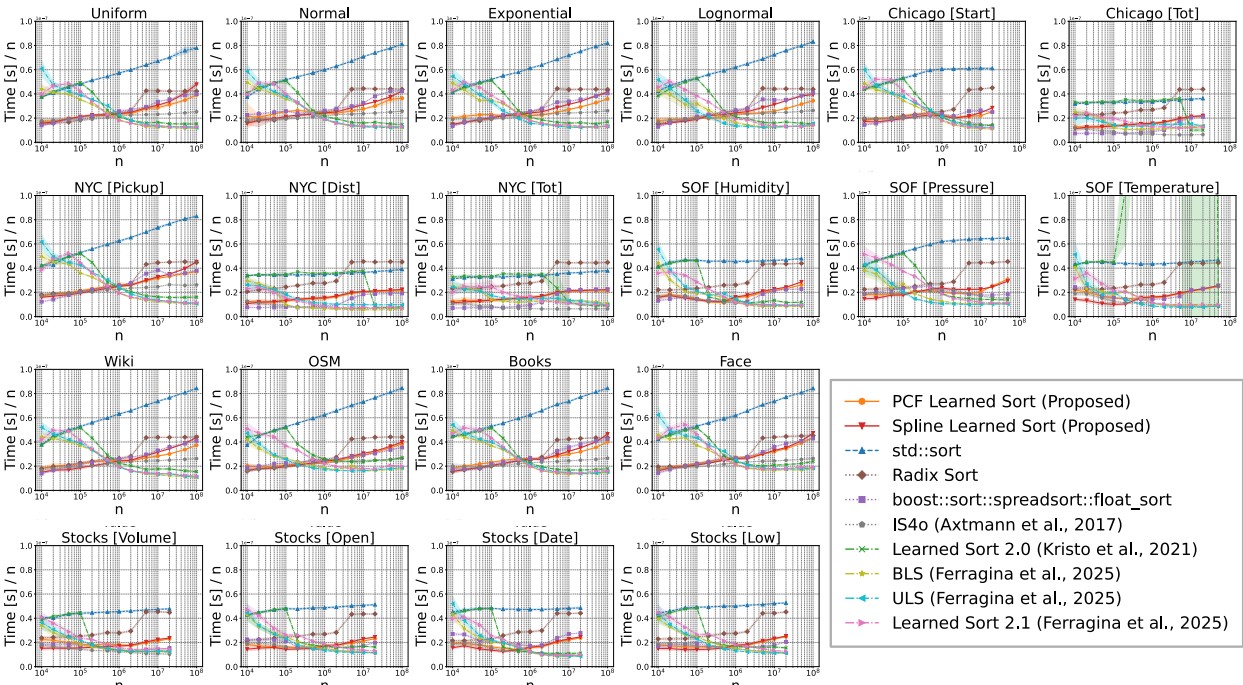

Figure 4: Time to sort the array. The standard deviation of the 10 measurements is represented by the shaded area. Our PCF Learned Sort is significantly faster than `std::sort`, and more importantly, it maintains robust performance across all datasets, whereas other learned sorts without worst-case guarantees can suffer catastrophic slowdowns, as seen with Learned Sort 2.0 on the SOF [Temperature] dataset.

`std::sort`, radix sort, `boost::sort::spreadsort::float_sort` (Boost C++ implementation of Spreadsort (Ross, 2002)), and Learned Sort 2.0 (Kristo et al., 2021b). In addition, we evaluated more recent state-of-the-art learned sorting algorithms (Ferragina & Odorisio, 2025)—Balanced Learned Sort (BLS), Unbalanced Learned Sort (ULS), and Learned Sort 2.1—as well as IS4o (Axtmann et al., 2017), one of the state-of-the-art non-learned sequential sample sort algorithms.

Figure 4 shows the sorting time divided by $n$. It shows the mean and standard deviation of the 10 measurements for each condition. Note that the horizontal axis is logarithmic, and therefore the curve of `std::sort`, which has a complexity of $\mathcal{O}(n \log n)$, is almost linear in this plot for synthetic datasets and real-world datasets with few duplicates. In contrast, our PCF Learned Sort graph shows a relatively slow increase, suggesting that PCF Learned Sort has a complexity much smaller than $\mathcal{O}(n \log n)$. We see that our PCF Learned Sort is up to 2.5 times faster than `std::sort`. Furthermore, we find that for relatively large $n$ ($> 10^6$), our PCF Learned Sort usually outperforms not only radix sort but also Spreadsort, an algorithm that cleverly incorporates the advantage of comparison sort into radix sort.

The figure also shows that highly optimized methods like IS4o and other learned sorts often outperform PCF Learned Sort in average speed. This is because these methods are highly optimized implementations that consider factors like cache efficiency, whereas our implementation prioritizes providing rigorous theoretical guarantees. However, this speed comes at the cost of robustness: learned sorts without worst-case guarantees can suffer catastrophic slowdowns. For example, on the SOF [Temperature] dataset with $n = 2 \times 10^7$, Learned Sort 2.0 took up to 326.4 seconds, while our method consistently finished in 0.45 seconds. Moreover, as shown in our adversarial analysis (Appendix C), other recent learned sorts (BLS, ULS, Learned Sort 2.1) also exhibit vulnerabilities. These results highlight the practical value of our theoretical guarantees.

To better understand runtime behavior, we profiled PCF Learned Sort across its stages (Appendix D). The results show that while the training stage is consistently lightweight across datasets and input sizes, the

bucketing stage often becomes the bottleneck. The dominant cost varies depending on both $n$ and the dataset characteristics, suggesting opportunities for further optimization.

## 5 Discussion

**Comparison with (Zeighami & Shahabi, 2024).** Most recently, a concurrent work (Zeighami & Shahabi, 2024) introduced a theoretical framework for learned database operations, including sorting, indexing, and cardinality estimation. A key strength of their approach is the formal definition of "distribution learnability" and its applicability to a broad class of distributions, including those subject to distribution shifts. Using this framework, they developed a Learned Sort algorithm with an expected running time of $O(n \log \log n)$ under certain distributional assumptions. While their approach employs a bucketing-based sorting algorithm similar to ours, there are several key differences between their method and ours.

First, their algorithm relies on detailed prior knowledge about the distribution; it explicitly requires a parameter $\varkappa_2$, which represents the "learning possibility" of the distribution (see Definition 3.2 in (Zeighami & Shahabi, 2024) for details). Since $\varkappa_2$ depends on $\rho_1$ and $\rho_2$, their algorithm cannot be executed without knowing the values of $\rho_1$ and $\rho_2$ (or at least their lower and upper bounds). In contrast, our algorithm does not require these values, making it more widely applicable.

Second, they do not provide experimental results. Since estimating $\rho_1$ and $\rho_2$ from real data is challenging, empirical evaluation of their method is difficult. On the other hand, since our algorithm does not depend on these specific values, it is easy to implement and evaluate experimentally.

Finally, their algorithm lacks a worst-case complexity guarantee. In particular, there are cases where the algorithm may not terminate, making it impossible to give an upper bound on its worst-case complexity. In contrast, we provide a formal worst-case complexity guarantee.

**Limitations of the Current Theoretical Framework.** Our proof of the $\mathcal{O}(n \log \log n)$ expected complexity (Theorem 3.6), does not extend to distributions with $f(x) = 0$ or $\infty$. A similar limitation is observed in Learned Indexes (Zeighami & Shahabi, 2023), and extending the theory to cover a broader class of distributions remains an open direction in both Learned Indexes and Learned Sorts contexts. One promising direction is to integrate more advanced CDF approximation methods with theoretical guarantees into our Learned Sort algorithm. By adopting a refined CDF model and a bucketing algorithm that satisfies the conditions of Lemma 3.3, it may be possible to achieve stronger theoretical guarantees.

**Implementation Considerations and Optimizations.** A straightforward implementation of PCF Learned Sort is not in-place, as it requires an auxiliary buffer for buckets nearly as large as the input. Engineering techniques from highly optimized sample-sort implementations (Axtmann et al., 2017; 2022) suggest in-place variants without changing the asymptotic structure of our algorithm. Another important aspect is real-world efficiency: optimizing memory access patterns to enhance cache efficiency, reducing cache misses, and leveraging cache-aware data layouts could improve empirical performance without compromising theoretical guarantees. Dynamically tuning parameters such as $\alpha$, $\beta$, $\gamma$, and $\delta$ based on the input distribution may yield further performance gains while maintaining guarantees.

**Parallelization Potential.** Although our analysis targets the sequential setting, the structure of PCF Learned Sort admits parallelization at several stages: training the CDF model, computing bucket IDs, scattering into buckets, and sorting buckets are all amenable to data parallelism. This observation is consistent with prior engineering work that integrates learned partitioning with high-performance (in-place) sample-sort pipelines and parallel learned-sorting frameworks (Axtmann et al., 2017; Carvalho, 2022; Carvalho & Lawrence, 2023). These results complement our theoretical analysis and suggest opportunities for developing parallel learned-sorting algorithms with provable guarantees.

# 6    Conclusion

We proposed PCF Learned Sort, which provides guarantees on both its expected and worst-case complexities. We then confirmed these computational complexities empirically on both synthetic and real data. This is the first study to support the empirical success of Learned Sort theoretically and provides insight into why Learned Sort is fast.

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

# A   Proofs

Here, we give the proofs omitted in the main paper. In Appendix A.1, we give the proof of Lemma 3.1, which is important for proving the worst-case complexity of PCF Learned Sort. Appendix A.2 and Appendix A.3 give proofs of Lemma 3.3 and Lemma 3.4, respectively, which are important for proving the expected computational complexity of PCF Learned Sort. The quantization-aware version of Lemma 3.4 and Theorem 3.6 is defined and the proof is given in the Appendix A.4. Finally, we provide the proofs of the theorems guaranteeing the complexity of Spline Learned Sort in Appendix A.5.

## A.1   Proof of Lemma 3.1

*Proof.* Let $P(n)$ be the worst-case complexity of our Learned Sort when using the model-based bucketing algorithm $\mathcal{M}$ as assumed in Lemma 3.1 and $\delta = \lfloor n^d \rfloor$. Let $S(n)$ be the worst-case complexity of the standard sort and $R(n)$ be the worst-case complexity of model-based bucketing (including model training and inferences). Since $S(n) = \mathcal{O}(nU(n))$ and the standard sort terminates after a finite number of operations,

$$\exists\, C_1, l_1\ (>0),\ \ n \geq 0\ \ \Rightarrow\ \ S(n) \leq C_1 + l_1 n U(n). \tag{3}$$

Since $R(n) = \mathcal{O}(n)$ and $\gamma + 1 = \mathcal{O}(n)$,

$$\exists\, n_2, l_2\ (>0),\ \ n \geq n_2\ \ \Rightarrow\ \ R(n) \leq l_2 n. \tag{4}$$

$$\exists\, n_3, l_3\ (>0),\ \ n \geq n_3\ \ \Rightarrow\ \ \gamma + 1 \leq l_3 n. \tag{5}$$

In the following, we prove $P(n) = \mathcal{O}(nU(n) + n \log \log n)$ by mathematical induction.

First, for $n < \max(n_2, n_3, \tau) =: n_0$, there exists a constant $C\ (>0)$ such that $P(n) \leq C$. That is, for $n < n_0$, our Learned Sort terminates in a finite number of operations. This is because, since $\delta < n$, the bucket will either be smaller than the original array length $n$, or the bucket will be immediately sorted by the standard sort.

Next, assume that there exists a constant $C\ (>0)$ and $l\ (>0)$ such that $P(n) \leq C + l(nU(n) + n \log \log n)$ for all $n < k$, where $k$ is an integer such that $k \geq n_0$. Without loss of generality, we assume that $C \geq C_1$ and $l \geq l_1$. Let $\mathcal{S}_{\gamma,k}$ be the set consisting of all $(\gamma + 1)$-dimensional vectors of positive integers whose sum is $k$, i.e., $\mathcal{S}_{\gamma,k} := \left\{ \boldsymbol{s} \in \mathbb{Z}_{\geq 0}^{\gamma} \mid \sum_{i=1}^{\gamma+1} s_i = k \right\}$. Then,

$$
\begin{aligned}
P(k) &\leq R(k) + \max_{\boldsymbol{s} \in \mathcal{S}_{\gamma,k}} \sum_{i=1}^{\gamma+1} \left\{ \mathbb{1}[s_i \geq \lfloor k^d \rfloor] \cdot S(s_i) + \mathbb{1}[s_i < \lfloor k^d \rfloor] \cdot P(s_i) \right\} \\
&\leq l_2 k + \max_{\boldsymbol{s} \in \mathcal{S}_{\gamma,k}} \sum_{i=1}^{\gamma+1} \left\{ \mathbb{1}[s_i \geq \lfloor k^d \rfloor] \cdot (C_1 + l_1 s_i U(s_i)) + \mathbb{1}[s_i < \lfloor k^d \rfloor] \cdot (C + l(s_i U(s_i) + s_i \log \log s_i)) \right\} \\
&\leq l_2 k + \max_{\boldsymbol{s} \in \mathcal{S}_{\gamma,k}} \sum_{i=1}^{\gamma+1} \left( C + l s_i U(s_i) + l s_i \log \log k^d \right) \\
&\leq l_2 k + C(\gamma + 1) + lk U(k) + lk \log \log k^d \\
&\leq l_2 k + C l_3 k + lk U(k) + lk \log \log k + lk \log d \\
&\leq \left( l_2 + C l_3 - l \log \frac{1}{d} \right) k + (C + l(k U(k) + k \log \log k)).
\end{aligned}
\tag{6}
$$

Therefore, if we take $l$ such that

$$\frac{l_2 + C l_3}{\log \frac{1}{d}} \leq l, \tag{7}$$

then $P(k) \leq C + l(k U(k) + k \log \log k)$ (note that the left side of Equation (7) is a constant independent of $k$).

Hence, by mathematical induction, it is proved that there exists a constant $C\ (>0)$ and $l\ (>0)$ such that $P(n) \leq C + l(k U(k) + lk \log \log k)$ for all $n \in \mathbb{N}$. $\qquad \square$

### A.2 Proof of Lemma 3.3

*Proof.* The proof approach is the same as in Lemma 3.1, but in Lemma 3.3, the "expected" computational complexity is bounded. The following two randomnesses are considered for computing the "expected" computational complexity: (i) the randomness with which $n$ elements are independently sampled according to the probability density function $f(x)$ in the process of forming the input array $\boldsymbol{x}$, and (ii) the randomness of the PCF Learned Sort algorithm sampling $\alpha$ elements from the input array $\boldsymbol{x}$ for training the PCF.

Let $T(n)$ be the expected complexity of our Learned Sort when using the model-based bucketing algorithm $\mathcal{M}$ as assumed in Lemma 3.3 and $\delta = \lfloor n^d \rfloor$. Let $S(n)$ be the expected complexity of the standard sort and $R(n)$ be the expected complexity of model-based bucketing (including model training and inferences). Since $S(n) = \mathcal{O}(n \log n)$ and the standard sort terminates after a finite number of operations,

$$\exists\, C_1, l_1\ (>0),\ \ n \geq 0\ \Rightarrow\ S(n) \leq C_1 + l_1 n \log n. \tag{8}$$

Since $R(n) = \mathcal{O}(n)$, $\Pr[\exists j, |\boldsymbol{c}_j| \geq \lfloor n^d \rfloor] = \mathcal{O}(1/\log n)$, and $\gamma + 1 = \mathcal{O}(n)$,

$$\exists\, n_2, l_2\ (>0),\ \ n \geq n_2\ \Rightarrow\ R(n) \leq l_2 n, \tag{9}$$

$$\exists\, n_3, l_3\ (>0),\ \ n \geq n_3\ \Rightarrow\ \Pr[\exists j, |\boldsymbol{c}_j| \geq \lfloor n^d \rfloor] \leq \frac{l_3}{\log n}, \tag{10}$$

$$\exists\, n_4, l_4\ (>0),\ \ n \geq n_4\ \Rightarrow\ \gamma + 1 \leq l_4 n. \tag{11}$$

In the following, we prove $T(n) = \mathcal{O}(n \log \log n)$ by mathematical induction.

First, for $n < \max(n_2, n_3, n_4, \tau) =: n_0$, there exists a constant $C\ (>0)$ such that $T(n) \leq C$. That is, for $n < n_0$, our Learned Sort terminates in a finite number of operations.

Next, assume that there exists a constant $C\ (>0)$ and $l\ (>0)$ such that $T(n) \leq C + ln \log \log n$ for all $n < k$, where $k$ is an integer such that $k \geq n_0$. Then, from $k \geq n_2$,

$$T(k) \leq R(k) + \mathbb{E}\left[ \sum_{j=1}^{\gamma+1} \mathbb{1}[|\boldsymbol{c}_j| \geq \lfloor k^d \rfloor] \cdot S(|\boldsymbol{c}_j|) + \mathbb{1}[|\boldsymbol{c}_j| < \lfloor k^d \rfloor] \cdot T(|\boldsymbol{c}_j|) \right]$$

$$\leq l_2 k + \Pr\left[\exists j, |\boldsymbol{c}_j| \geq \lfloor k^d \rfloor\right] \cdot \mathbb{E}\left[ \sum_{j=1}^{\gamma+1} S(|\boldsymbol{c}_j|) \right] + \mathbb{E}\left[ \sum_{j=1}^{\gamma+1} \mathbb{1}[|\boldsymbol{c}_j| < \lfloor k^d \rfloor] \cdot T(|\boldsymbol{c}_j|) \right] \tag{12}$$

$$\leq l_2 k + \Pr[\exists j, |\boldsymbol{c}_j| \geq \lfloor k^d \rfloor] \cdot \{C_1(\gamma + 1) + l_1 k \log k\} + \mathbb{E}\left[ \sum_{i=1}^{\gamma+1} T(\min(\lfloor k^d \rfloor, |\boldsymbol{c}_j|)) \right].$$

Here, from $k \geq n_3$ and $k \geq n_4$,

$$\Pr[\exists j, |\boldsymbol{c}_j| \geq \lfloor k^d \rfloor] \cdot \{C_1(\gamma + 1) + l_1 k \log k\} \leq \frac{l_3}{\log k} \cdot (C_1 l_4 k + l_1 k \log k)$$

$$\leq (C_1 l_3 l_4 + l_1 l_3) k. \tag{13}$$

Also, from the assumption of induction and $k \geq n_4$,

$$\mathbb{E}\left[ \sum_{i=1}^{\gamma+1} T(\min(\lfloor k^d \rfloor, |\boldsymbol{c}_j|)) \right] \leq \mathbb{E}\left[ \sum_{i=1}^{\gamma+1} \{C + l \cdot \min(\lfloor k^d \rfloor, |\boldsymbol{c}_j|) \log \log \min(\lfloor k^d \rfloor, |\boldsymbol{c}_j|)\} \right]$$

$$\leq \mathbb{E}\left[ C(\gamma + 1) + \sum_{i=1}^{\gamma+1} l \cdot |\boldsymbol{c}_j| \log \log \lfloor k^d \rfloor \right] \tag{14}$$

$$\leq C l_4 k + l k \log \log \lfloor k^d \rfloor$$

$$\leq C l_4 k + l k \log d + l k \log \log k.$$

Therefore,

$$
\begin{aligned}
T(k) &\leq l_2 k + (C_1 l_3 l_4 + l_1 l_3)k + C l_4 k + lk \log d + lk \log \log k \\
&\leq \left\{ l_2 + C_1 l_3 l_4 + l_1 l_3 + C l_4 - l \log \frac{1}{d} \right\} k + (C + lk \log \log k).
\end{aligned}
\tag{15}
$$

Therefore, if we take $l$ such that

$$
\frac{l_2 + C_1 l_3 l_4 + l_1 l_3 + C l_4}{\log \frac{1}{d}} \leq l,
\tag{16}
$$

then $T(k) \leq C + lk \log \log k$ (note that the left side of Equation (16) is a constant independent of $k$).

Hence, by mathematical induction, it is proved that there exists a constant $C$ $(> 0)$ and $l$ $(> 0)$ such that $T(n) \leq C + ln \log \log n$ for all $n \in \mathbb{N}$. $\qquad\square$

## A.3 Proof of Lemma 3.4

We first present the following lemma to prove Lemma 3.4.

**Lemma A.1.** *Let $\boldsymbol{e}$ $(\in \mathcal{I}^n)$ be a sorted version of $\boldsymbol{x}_{\mathcal{I}}$ $(\in \mathcal{I}^n)$ and $\Delta := (x_{\max} - x_{\min})/\beta$ (where $x_{\min}$ and $x_{\max}$ are the minimum and maximum values of $\boldsymbol{x}_{\mathcal{I}}$, respectively).*

*Also, define the set $\mathcal{S}_r$ and $\mathcal{T}_r$ as follows $(r = 1, \ldots, n)$:*

$$
\mathcal{S}_r = \{k \mid e_{\max(1, r - \delta/2)} + \Delta < e_k \leq e_r - \Delta\}, \qquad \mathcal{T}_r = \{k \mid e_r + \Delta \leq e_k < e_{\min(r + \delta/2, n)} - \Delta\}.
\tag{17}
$$

*Using this definition, define $Y_{jr}, Z_{jr}, Y_r, Z_r$ as follows $(j = 1, \ldots, \alpha, \ r = 1, \ldots, n)$:*

$$
Y_{jr} = \begin{cases} 1 & (j \in \mathcal{S}_r) \\ 0 & (\text{else}) \end{cases}, \qquad
Z_{jr} = \begin{cases} 1 & (j \in \mathcal{T}_r) \\ 0 & (\text{else}) \end{cases}, \qquad
Y_r = \sum_{j=1}^{\alpha} Y_{jr}, \qquad
Z_r = \sum_{j=1}^{\alpha} Z_{jr}
\tag{18}
$$

*When using $\mathcal{M}_{\mathrm{PCF}}$, if the size of the bucket to which $e_r$ is allocated is greater than or equal to $\delta$, the following holds:*

$$
\left( r \geq \frac{\delta}{2} + 1 \ \wedge \ Y_r \leq \left\lfloor \frac{\alpha}{\gamma} \right\rfloor \right) \ \vee \ \left( r \leq n - \frac{\delta}{2} \ \wedge \ Z_r \leq \left\lfloor \frac{\alpha}{\gamma} \right\rfloor \right).
\tag{19}
$$

*Proof.* We prove the contraposition of the lemma. That is, we prove that $e_r$ is allocated to a bucket smaller than $\delta$ under the assumption that $\left( r < \frac{\delta}{2} + 1 \ \vee \ Y_r > \left\lfloor \frac{\alpha}{\gamma} \right\rfloor \right) \ \wedge \ \left( r > n - \frac{\delta}{2} \ \vee \ Z_r > \left\lfloor \frac{\alpha}{\gamma} \right\rfloor \right)$.

For convenience, we hypothetically define $e_0 = -\infty, e_{n+1} = \infty$, and assign $e_0$ to the 0th bucket and $e_{n+1}$ to the $(\gamma + 2)$-th bucket. The size of the 0th bucket and the $(\gamma + 2)$-th bucket are both 1.

First, we prove that $e_r$ and $e_{\min(r + \delta/2, n+1)}$ are assigned to different buckets. When $n - \delta/2 < r \leq n$, $e_r$ and $e_{\min(r + \delta/2, n+1)} = e_{n+1}$ are obviously assigned to different buckets. When $r \leq n - \delta/2$, the ID of the bucket to which $e_r$ is assigned is

$$
\begin{aligned}
\lfloor \tilde{F}(e_r)\gamma \rfloor + 1 &= \left\lfloor \frac{\gamma}{\alpha} b_{i(e_r)} \right\rfloor + 1 \\
&\leq \frac{\gamma}{\alpha} b_{i(e_r)} + 1 \\
&= \frac{\gamma}{\alpha} |\{j \mid i(a_j) \leq i(e_r)\}| + 1 \\
&\leq \frac{\gamma}{\alpha} |\{j \mid a_j \leq e_r + \Delta\}| + 1.
\end{aligned}
\tag{20}
$$

The ID of the bucket to which $e_{\min(r+\delta/2,n+1)} = e_{r+\delta/2}$ is assigned is

$$
\begin{aligned}
\lfloor \tilde{F}(e_{r+\delta/2})\gamma \rfloor + 1 &= \left\lfloor \frac{\gamma}{\alpha} b_{i(e_{r+\delta/2})} \right\rfloor + 1 \\
&> \frac{\gamma}{\alpha} b_{i(e_{r+\delta/2})} \\
&= \frac{\gamma}{\alpha} \left| \{j \mid i(a_j) \le i(e_{r+\delta/2})\} \right| \\
&\ge \frac{\gamma}{\alpha} \left| \{j \mid a_j \le e_{r+\delta/2} - \Delta\} \right|.
\end{aligned}
\tag{21}
$$

Thus, taking the difference between these two bucket IDs,

$$
\begin{aligned}
\left( \lfloor \tilde{F}(e_{r+\delta/2})\gamma \rfloor + 1 \right) - \left( \lfloor \tilde{F}(e_r)\gamma \rfloor + 1 \right) &> \frac{\gamma}{\alpha} \left| \{j \mid a_j \le e_{r+\delta/2} - \Delta\} \right| - \left( \frac{\gamma}{\alpha} \left| \{j \mid a_j \le e_r + \Delta\} \right| + 1 \right) \\
&= \frac{\gamma}{\alpha} \left| \{j \mid e_r + \Delta < a_j \le e_{r+\delta/2} - \Delta\} \right| - 1 \\
&= \frac{\gamma}{\alpha} |\mathcal{T}_r| - 1 \\
&= \frac{\gamma}{\alpha} \sum_{j=1}^{\alpha} Z_{jr} - 1 \\
&= \frac{\gamma}{\alpha} Z_r - 1 \\
&\ge 0.
\end{aligned}
\tag{22}
$$

Therefore, $e_r$ and $e_{\min(r+\delta/2,n+1)}$ are assigned to different buckets.

In the same way, we can prove that $e_{\max(0,r-\delta/2)}$ and $e_r$ are also assigned to different buckets. Thus, the size of the bucket to which $e_r$ is assigned is at most $\delta - 1$ (at most from $e_{\max(0,r-\delta/2)+1}$ to $e_{\min(r+\delta/2,n+1)-1}$), and the contraposition of the lemma is proved. $\qquad\square$

Using Lemma A.1, we can prove Lemma 3.4.

*Proof.* Let $q = \max_y \int_y^{y+\Delta} f_{\mathcal{I}}(x)dx$ (where $y$ is a value such that $(y, y+\Delta) \subseteq \mathcal{I}$). Then, from $\sigma_1 \le f(x) \le \sigma_2$ for all $x \in \mathcal{I}$,

$$
\begin{aligned}
q &\le \frac{\max_y \int_y^{y+\Delta} f(y)dy}{\int_{\mathcal{I}} f(x)dx} \\
&\le \frac{\max_y \int_y^{y+\Delta} \sigma_2 dy}{\int_{\mathcal{I}} \sigma_1 dx} \\
&\le \frac{\sigma_2 \Delta}{\sigma_1 (x_{\max} - x_{\min})} \\
&= \frac{\sigma_2}{\sigma_1 \beta}.
\end{aligned}
\tag{23}
$$

Thus, when $r \ge \frac{\delta}{2} + 1$,

$$
\begin{aligned}
\mathbb{E}\left[ \frac{\delta}{2} - |\mathcal{S}_r| \right] &= \mathbb{E}\left[ \frac{\delta}{2} - \left| \{k \mid e_{r-\delta/2} + \Delta < e_k \le e_r - \Delta\} \right| \right] \\
&= \mathbb{E}\left[ \left| \{k \mid e_{r-\delta/2} < e_k \le e_{r-\delta/2} + \Delta\} \right| \right] + \mathbb{E}\left[ \left| \{k \mid e_r - \Delta < e_k \le e_r\} \right| \right] \\
&\le nq + nq \\
&\le \frac{2\sigma_2 n}{\sigma_1 \beta}.
\end{aligned}
\tag{24}
$$

Thus, when $r \geq \frac{\delta}{2} + 1$,

$$
\begin{aligned}
\mathbb{E}[Y_r] &= \frac{\alpha}{n} \mathbb{E}\left[|\mathcal{S}_r|\right] \\
&= \frac{\alpha}{n}\left(\frac{\delta}{2} - \mathbb{E}\left[\frac{\delta}{2} - |\mathcal{S}_r|\right]\right) \\
&\geq \frac{\alpha\delta}{2n} - \frac{2\sigma_2\alpha}{\sigma_1\beta} \\
&= \frac{\alpha K}{\gamma}.
\end{aligned}
\tag{25}
$$

Here, when $K \geq 1$, we have

$$
0 \leq 1 - \frac{\alpha}{\gamma\mathbb{E}[Y_r]} < 1.
\tag{26}
$$

Therefore, from the Chernoff bound,

$$
\begin{aligned}
\Pr\left[Y_r \leq \frac{\alpha}{\gamma}\right] &= \Pr\left[Y_r \leq \left\{1 - \left(1 - \frac{\alpha}{\gamma\mathbb{E}[Y_r]}\right)\right\}\mathbb{E}\left[Y_r\right]\right] \\
&\leq \exp\left\{-\frac{1}{2}\left(1 - \frac{\alpha}{\gamma\mathbb{E}[Y_r]}\right)^2 \mathbb{E}\left[Y_r\right]\right\} \\
&\leq \exp\left\{-\frac{\alpha K}{2\gamma}\left(1 - \frac{1}{K}\right)^2\right\}.
\end{aligned}
\tag{27}
$$

In the same way, we can prove that when $r \leq n - \frac{\delta}{2}$,

$$
\Pr\left[Z_r \leq \frac{\alpha}{\gamma}\right] \leq \exp\left\{-\frac{\alpha K}{2\gamma}\left(1 - \frac{1}{K}\right)^2\right\}.
\tag{28}
$$

Thus, by defining $E_r$ to be the event "$e_r$ is allocated to a bucket with size greater than or equal to $\delta$," from Lemma A.1,

$$
\begin{aligned}
\Pr[E_r] &\leq \Pr\left[\left(r \geq \frac{\delta}{2} + 1 \ \wedge \ Y_r \leq \left\lfloor\frac{\alpha}{\gamma}\right\rfloor\right) \vee \left(r \leq n - \frac{\delta}{2} \ \wedge \ Z_r \leq \left\lfloor\frac{\alpha}{\gamma}\right\rfloor\right)\right] \\
&\leq \Pr\left[\left(r \geq \frac{\delta}{2} + 1 \ \wedge \ Y_r \leq \left\lfloor\frac{\alpha}{\gamma}\right\rfloor\right)\right] + \Pr\left[\left(r \leq n - \frac{\delta}{2} \ \wedge \ Z_r \leq \left\lfloor\frac{\alpha}{\gamma}\right\rfloor\right)\right] \\
&\leq \begin{cases} \exp\left\{-\frac{\alpha K}{2\gamma}\left(1 - \frac{1}{K}\right)^2\right\}, & \left(r < \frac{\delta}{2} + 1 \ \vee \ r > n - \frac{\delta}{2}\right) \\ 2\exp\left\{-\frac{\alpha K}{2\gamma}\left(1 - \frac{1}{K}\right)^2\right\}, & \text{(else)} \end{cases} \\
&\leq 2\exp\left\{-\frac{\alpha K}{2\gamma}\left(1 - \frac{1}{K}\right)^2\right\}.
\end{aligned}
\tag{29}
$$

Therefore,

$$
\mathbb{E}\left[\sum_{r=1}^{n} \mathbb{1}[E_r]\right] \leq 2n\exp\left\{-\frac{\alpha K}{2\gamma}\left(1 - \frac{1}{K}\right)^2\right\}
\tag{30}
$$

Then, noting that the number of buckets with size greater than or equal to $\delta$ is less than or equal to $\sum_{r=1}^{n} \mathbb{1}[E_r]/\delta$,

$$
\mathbb{E}\left[|\{j \mid |\boldsymbol{c}_j| > \delta\}|\right] \leq \frac{2n}{\delta}\exp\left\{-\frac{\alpha K}{2\gamma}\left(1 - \frac{1}{K}\right)^2\right\}.
\tag{31}
$$

Then, from Markov's inequality, we have

$$
\begin{aligned}
\Pr[\exists j, |\boldsymbol{c}_j| > \delta] &= \Pr[|\{j \mid |\boldsymbol{c}_j| > \delta\}| \geq 1] \\
&\leq \frac{2n}{\delta} \exp\left\{ -\frac{\alpha K}{2\gamma} \left(1 - \frac{1}{K}\right)^2 \right\}.
\end{aligned}
\tag{32}
$$

$\square$

### A.4 The Quantization-Aware Version of Lemma 3.4 and Theorem 3.6

In general, computers represent numbers in a finite number of bits, so the numbers they handle are inherently discrete. However, Theorem 3.6, which states that the expected computational complexity of PCF Learned Sort is $\mathcal{O}(n \log \log n)$, does not cover discrete distributions. Here, we define the quantization process by which a computer represents numbers in finite bits and then show that the expected computational complexity of PCF Learned Sort is still $\mathcal{O}(n \log \log n)$, under the assumption that "the quantization is fine enough."

First, we assume the sampling and quantization process is as follows:

**Assumption A.2.** For a range of values $\mathcal{D}$ ($\subseteq \mathbb{R}$), define $m$ ($\in \mathbb{N}$) contiguous regions $\mathcal{D}_1, \mathcal{D}_2, \ldots, \mathcal{D}_m$ such that (i) they are disjoint from each other and (ii) together they form $\mathcal{D}$. For each region, determine representative values $r_1, r_2, \ldots, r_m$. Here, $r_1, r_2, \ldots, r_m$ are values contained in $\mathcal{D}_1, \mathcal{D}_2, \ldots, \mathcal{D}_m$, respectively. For a value $x$ ($\in \mathcal{D}$), the quantized value of $x$, $x'$, is obtained as $x' = r_i$, where $i$ is the (only) $i$ such that $x \in \mathcal{D}_i$. The value $x$ is sampled according to the probability density function $f(x)$, but a computer keeps $x'$ (instead of $x$) in $\log_2 m$ bits with some quantization error.

Also, for the interval $\mathcal{I} \subseteq \mathcal{D}$, we define $\eta(\mathcal{I})$ as follows.

**Definition A.3.** $\eta(\mathcal{I})$ is the maximum width of $\mathcal{D}_i$ that intersects with interval $\mathcal{I}$, i.e.,

$$
\eta(\mathcal{I}) \coloneqq \max\left\{ |\mathcal{D}_i| \mid \mathcal{D}_i \cup \mathcal{I} \neq \varnothing, i = 1, \ldots, m \right\},
\tag{33}
$$

where $|\mathcal{D}_i|$ is the width of the range $\mathcal{D}_i$.

We can prove that $\eta(\mathcal{I})$ is the upper bound of the quantization error of $x$ in $\mathcal{I}$, i.e., $|x - x'| \leq \eta(\mathcal{I})$ when $x \in \mathcal{I}$.

Now, the assumption that the quantization is "fine enough" is specifically defined as follows.

**Assumption A.4.** (Recall that our PCF Learned Sort recursively calls its own algorithm. Each time of recursion, the range of values of interest $\mathcal{I}$ changes, and the length of the array of interest $n$ also changes.) For all $\mathcal{I}$ and $n$ that appear in the algorithm, the following holds:

$$
\frac{\beta \eta(\mathcal{I})}{|\mathcal{I}|} \leq \frac{1}{2}.
\tag{34}
$$

We show intuitively and empirically that this is a satisfactory assumption.

First, to show intuitively that this assumption is satisfactory, we give an example. Let $\mathcal{I}$ be a range of values that can be represented by a 64-bit double (1 bit for the sign, 11 bits for the exponent part, and 52 bits for the mantissa part), that is, $\mathcal{I} = [-1.79 \times 10^{308}, 1.79 \times 10^{308}]$. The quantization is performed by mapping each value to the nearest number that can be represented by a 64-bit double. In this setting, $\eta(\mathcal{I}) = 1.99 \times 10^{292}$, $|\mathcal{I}| = 3.59 \times 10^{308}$. $\eta(\mathcal{I})/|\mathcal{I}| = 5.55 \times 10^{-17}$. Thus, for usual $\beta$, $\beta \leq 9 \times 10^{15}$, Equation (34) holds.

Second, we show that Equation (34) is a satisfactory assumption empirically. In the 1,280 measurements, where 10 measurements each for 16 different $n(\in \{10^3, 2 \times 10^3, 5 \times 10^3, \ldots, 10^8\})$ on 8 different datasets, $5.23 \times 10^7$ pairs of $(\mathcal{I}, \beta)$ appeared (we set $\beta = \lfloor n^{3/4} \rfloor$), and the left side of Equation (34) is at most $8.11 \times 10^{-9}$, indicating that Equation (34) is always true with a margin.

Under this definition and assumption about quantization, we can prove the quantization-aware version of Lemma 3.4.

**Lemma A.5** (Quantization-aware version of Lemma 3.4)**.** *Let $\sigma_1$ and $\sigma_2$ be respectively the lower and upper bounds of the probability density distribution $f(x)$ in $\mathcal{D}$, and assume that $0 < \sigma_1 \le \sigma_2 < \infty$. That is, $x \in \mathcal{D} \Rightarrow \sigma_1 \le f(x) \le \sigma_2$. Also, let $\boldsymbol{x}'_{\mathcal{I}}$ be the array created by the quantization of $\boldsymbol{x} \; (\in \mathcal{I}^n)$ in the manner defined in Assumption A.2.*

*Then, in model-based bucketing of $\boldsymbol{x}'_{\mathcal{I}} \; (\in \mathcal{I}^n)$ to $\{\boldsymbol{c}_j\}_{j=1}^{\gamma+1}$ using $\mathcal{M}_{\mathrm{PCF}}$, the following holds for any interval $\mathcal{I} \; (\subseteq \mathcal{D})$ under Assumption A.4:*

$$K := \frac{\gamma \delta}{2n} - \frac{4\sigma_2 \gamma}{\sigma_1 \beta} \ge 1 \;\; \Rightarrow \;\; \Pr[\exists j, |\boldsymbol{c}_j| > \delta] \le \frac{2n}{\delta} \exp\left\{-\frac{\alpha K'}{2\gamma}\left(1 - \frac{1}{K'}\right)^2\right\}. \tag{35}$$

The proof of Lemma A.5 is done in the same way as Lemma 3.4. That is, we first prove the following lemma.

**Lemma A.6** (Quantization-aware version of Lemma A.6)**.** *Let $\boldsymbol{e}' \; (\in \mathcal{I}^n)$ be a sorted version of $\boldsymbol{x}'_{\mathcal{I}} \; (\in \mathcal{I}^n)$ and $\Delta := (x'_{\max} - x'_{\min})/\beta$ (where $x'_{\min}$ and $x'_{\max}$ are the minimum and maximum values of $\boldsymbol{x}'_{\mathcal{I}}$, respectively).*

*Also, define the set $\mathcal{S}'_r, \mathcal{T}'_r$ as follows ($r = 1, \ldots, n$):*

$$\mathcal{S}'_r = \{k \mid e_{\max(1, r-\delta/2)} + \Delta + 2\eta(\mathcal{I}) < e_k \le e_r - \Delta - 2\eta(\mathcal{I})\}, \tag{36}$$

$$\mathcal{T}'_r = \{k \mid e_r + \Delta + 2\eta(\mathcal{I}) \le e_k < e_{\min(r+\delta/2, n)} - \Delta - 2\eta(\mathcal{I})\}. \tag{37}$$

*Using this definition, define $Y'_{jr}, Z'_{jr}, Y'_r, Z'_r$ as follows ($j = 1, \ldots, \alpha, \; r = 1, \ldots, n$):*

$$Y'_{jr} = \begin{cases} 1 & (j \in \mathcal{S}'_r) \\ 0 & (\text{else}) \end{cases}, \qquad Z'_{jr} = \begin{cases} 1 & (j \in \mathcal{T}'_r) \\ 0 & (\text{else}) \end{cases}, \qquad Y'_r = \sum_{j=1}^{\alpha} Y'_{jr}, \qquad Z'_r = \sum_{j=1}^{\alpha} Z'_{jr}. \tag{38}$$

*If the size of the bucket to which $e'_r$ is allocated is greater than or equal to $\delta$, then the following holds:*

$$\left(r \ge \frac{\delta}{2} + 1 \;\wedge\; Y'_r \le \left\lfloor \frac{\alpha}{\gamma} \right\rfloor\right) \;\vee\; \left(r \le n - \frac{\delta}{2} \;\wedge\; Z'_r \le \left\lfloor \frac{\alpha}{\gamma} \right\rfloor\right). \tag{39}$$

*Proof.* The proof method is exactly the same as for Lemma A.1. That is, by taking the difference between the IDs of the buckets to which $e'_{r+\delta/2}$ and $e'_r$ are assigned,

$$\begin{aligned}
&\left(\lfloor \tilde{F}(e'_{r+\delta/2})\gamma \rfloor + 1\right) - \left(\lfloor \tilde{F}(e'_r)\gamma \rfloor + 1\right) \\
&> \frac{\gamma}{\alpha} \left|\{j \mid a_j \le e_{r+\delta/2} - \Delta - 2\eta(\mathcal{I})\}\right| - \left(\frac{\gamma}{\alpha} \left|\{j \mid a_j \le e_r + \Delta + 2\eta(\mathcal{I})\}\right| + 1\right) \\
&= \frac{\gamma}{\alpha} \left|\{j \mid e_r + \Delta + 2\eta(\mathcal{I}) < a_j \le e_{r+\delta/2} - \Delta - 2\eta(\mathcal{I})\}\right| - 1 \\
&= \frac{\gamma}{\alpha} Z'_r - 1 \\
&\ge 0,
\end{aligned} \tag{40}$$

when $Z'_r > \left\lfloor \frac{\alpha}{\gamma} \right\rfloor$. Thus, we can prove that when

$$\left(r < \frac{\delta}{2} + 1 \;\vee\; Y'_r > \left\lfloor \frac{\alpha}{\gamma} \right\rfloor\right) \;\wedge\; \left(r > n - \frac{\delta}{2} \;\vee\; Z'_r > \left\lfloor \frac{\alpha}{\gamma} \right\rfloor\right) \tag{41}$$

holds, $e'_{r+\delta/2}$ and $e'_r$ are assigned to the different bucket and $e'_{r-\delta/2}$ and $e'_r$ are assigned to the different bucket. Then, the contraposition of the lemma is proved. $\square$

Using Lemma A.6, we can prove Lemma A.5.

*Proof.* Let $q' = \max_y \int_y^{y+\Delta 2\eta(\mathcal{I})} f_{\mathcal{I}}(x)dx$ (where $y$ is a value such that $(y, y + \Delta + 2\eta(\mathcal{I}) \subseteq \mathcal{I})$. Then, from $\sigma_1 \le f(x) \le \sigma_2$ for all $x \in \mathcal{I}$,

$$
\begin{aligned}
q' &\le \frac{\max_y \int_y^{y+\Delta+2\eta(\mathcal{I})} f(y)dy}{\int_{\mathcal{I}} f(x)dx} \\
&\le \frac{\max_y \int_y^{y+\Delta+2\eta(\mathcal{I})} \sigma_2 dy}{\int_{\mathcal{I}} \sigma_1 dx} \\
&= \frac{\sigma_2(\Delta + 2\eta(\mathcal{I}))}{\sigma_1(x'_{\max} - x'_{\min})} \\
&\le \frac{\sigma_2}{\sigma_1 \beta} \cdot \left(1 + \frac{2\beta\eta(\mathcal{I})}{|\mathcal{I}|}\right) \\
&\le \frac{2\sigma_2}{\sigma_1 \beta}.
\end{aligned}
\tag{42}
$$

The last inequality is obtained by Assumption A.4.

Thus, when $r \ge \frac{\delta}{2} + 1$,

$$
\begin{aligned}
\mathbb{E}\left[\frac{\delta}{2} - |\mathcal{S}'_r|\right] &= \mathbb{E}\left[\frac{\delta}{2} - \left|\{k \mid e_{r-\delta/2} + \Delta + 2\eta(\mathcal{I}) < e_k \le e_r - \Delta - 2\eta(\mathcal{I})\}\right|\right] \\
&= \mathbb{E}\left[\left|\{k \mid e_{r-\delta/2} < e_k \le e_{r-\delta/2} + \Delta + 2\eta(\mathcal{I})\}\right|\right] + \mathbb{E}\left[\left|\{k \mid e_r - \Delta - 2\eta(\mathcal{I}) < e_k \le e_r\}\right|\right] \\
&\le nq' + nq' \\
&\le \frac{4\sigma_2 n}{\sigma_1 \beta}.
\end{aligned}
\tag{43}
$$

Thus, when $r \ge \frac{\delta}{2} + 1$,

$$
\begin{aligned}
\mathbb{E}[Y'_r] &= \frac{\alpha}{n}\mathbb{E}\left[|\mathcal{S}'_r|\right] \\
&= \frac{\alpha}{n}\left(\frac{\delta}{2} - \mathbb{E}\left[\frac{\delta}{2} - |\mathcal{S}'_r|\right]\right) \\
&\ge \frac{\alpha\delta}{2n} - \frac{4\sigma_2\alpha}{\sigma_1\beta} \\
&= \frac{\alpha K'}{\gamma}.
\end{aligned}
\tag{44}
$$

From this point forward, by proceeding in exactly the same way as the proof of Lemma 3.4, we can prove the following using Lemma A.1:

$$
K' \ge 1 \quad \Rightarrow \quad \Pr[\exists j, |c_j| > \delta] \le \frac{2n}{\delta}\exp\left\{-\frac{\alpha K'}{2\gamma}\left(1 - \frac{1}{K'}\right)^2\right\}.
\tag{45}
$$

$\square$

Using Lemma A.5, we can prove the following theorem.

**Theorem A.7** (Quantization-aware version of Theorem 3.6)**.** *Let $\sigma_1$ and $\sigma_2$ be the lower and upper bounds, respectively, of the probability density distribution $f(x)$ in $\mathcal{D}$, and assume that $0 < \sigma_1 \le \sigma_2 < \infty$. Also, assume that the input array is quantized in a way that satisfies Assumption A.2 and Assumption A.4. Then, the expected complexity of PCF Learned Sort with $\mathcal{M}_{PCF}$ as the bucketing method and $\alpha = \beta = \gamma = \delta = \lfloor n^{3/4} \rfloor$ is $\mathcal{O}(n \log \log n)$.*

*Proof.* When $\alpha = \beta = \gamma = \lfloor n^{3/4} \rfloor$, the computational complexity for model-based bucketing is $\mathcal{O}(n)$. Since $K' = \Omega(\sqrt{n})$ when $\alpha = \beta = \gamma = \delta = \lfloor n^{3/4} \rfloor$, $K' \geq 1$ for sufficiently large $n$, and

$$\frac{2n}{\delta} \exp\left\{ -\frac{\alpha K'}{2\gamma} \left(1 - \frac{1}{K'}\right)^2 \right\} = \mathcal{O}(n^{\frac{1}{4}} \exp(-\sqrt{n})) \leq \mathcal{O}\left(\frac{1}{\log n}\right). \tag{46}$$

Therefore, from Lemma 3.3 and Lemma A.5, the expected computational complexity of PCF Learned Sort is $\mathcal{O}(n \log \log n)$. $\qquad\square$

## A.5 Proofs for Spline Learned Sort

**Definition of Spline Learned Sort.** We now formally define *Spline Learned Sort*, which we propose as a variant of PCF Learned Sort. In particular, Spline Learned Sort replaces $\mathcal{M}_{\mathrm{PCF}}$—the bucketing method that uses PCF as the CDF model—with $\mathcal{M}_{\mathrm{Spline}}$, a bucketing method based on a spline-based CDF model. Below, we give a rigorous description of $\mathcal{M}_{\mathrm{Spline}}$.

The training algorithm of the CDF model in $\mathcal{M}_{\mathrm{Spline}}$ is the same as in $\mathcal{M}_{\mathrm{PCF}}$. That is, the parameters $\alpha, \beta \in \mathbb{N}$ are determined by $n$. Then, following this setting, we sample an array $\boldsymbol{a}$ of length $\alpha$ from the input array $\boldsymbol{x}$, which is subsequently bin-counted (as described in Section 3.3).

For later explanation and proofs, we introduce additional terminology. We define the vector of thresholds used for counting as $\boldsymbol{t} \in \mathbb{R}^{\beta+2}$: $t_i = x_{\min} + (x_{\max} - x_{\min})(i-1)/\beta$ for $i = 1, 2, \ldots, \beta + 2$. Thus, the $i$-th bin during CDF model training corresponds to the interval $[t_i, t_{i+1})$. Accordingly, the non-decreasing, non-negative array $\boldsymbol{b}$ formed by bin-counting is given by $b_i = |\{j \in \{1, \ldots, \alpha\} \mid a_j < t_{i+1}\}|$ for $i = 1, 2, \ldots, \beta + 1$. We also denote by $F_{\boldsymbol{a}}(x)$ the empirical CDF of the sampled array $\boldsymbol{a}$: $F_{\boldsymbol{a}}(x) = |\{j \in \{1, 2, \ldots, \alpha\} \mid a_j \leq x\}| / \alpha$.

The inference algorithm of the CDF model in $\mathcal{M}_{\mathrm{Spline}}$ is similar to but slightly different from that in $\mathcal{M}_{\mathrm{PCF}}$. In PCF, the prediction is constant within each interval $[t_i, t_{i+1})$. In contrast, in the spline-based case, the prediction is obtained by linearly interpolating the empirical CDF values at the endpoints of the interval. Specifically, the output $\tilde{F}(x)$ for input $x$ is obtained as follows. As in PCF, we compute $i(x) = \left\lfloor \frac{(x - x_{\min})\beta}{x_{\max} - x_{\min}} \right\rfloor + 1$, so that $x \in [t_{i(x)}, t_{i(x)+1})$. While PCF returns $F_{\boldsymbol{a}}(t_{i(x)+1}) = b_{i(x)}/\alpha$ as the CDF prediction, the spline-based method linearly interpolates between $F_{\boldsymbol{a}}(t_{i(x)})$ and $F_{\boldsymbol{a}}(t_{i(x)+1})$:

$$\tilde{F}(x) = F_{\boldsymbol{a}}(t_{i(x)}) + \frac{x - t_{i(x)}}{t_{i(x)+1} - t_{i(x)}} \cdot (F_{\boldsymbol{a}}(t_{i(x)+1}) - F_{\boldsymbol{a}}(t_{i(x)})) \tag{47}$$

$$= \frac{1}{\alpha} \left( b_{i(x)-1} + \frac{x - t_{i(x)}}{t_{i(x)+1} - t_{i(x)}} \cdot (b_{i(x)} - b_{i(x)-1}) \right), \tag{48}$$

with $b_0 = 0$.

Unlike PCF, which only considers which bin a value falls into, the spline-based method also accounts for the relative position of the value within the bin to compute the predicted CDF $\tilde{F}(x)$. The asymptotic computational complexity for training and inference is the same as PCF: $\mathcal{O}(n)$ for training and $\mathcal{O}(1)$ per element for inference. However, unlike PCF (which only requires array lookups), the spline-based approach requires additional subtractions and multiplications during inference, leading to a larger constant factor in runtime.

**Theorems Guaranteeing the Complexity of Spline Learned Sort.** When $\mathcal{M}_{\mathrm{Spline}}$ is used as the bucketing method in our learned sort framework, the same complexity guarantees as in PCF Learned Sort hold. In particular, we obtain the following worst-case guarantee (Theorem A.8) and expected-case guarantee (Theorem A.9).

**Theorem A.8** (Spline version of Theorem 3.5)**.** *If $\mathcal{M}_{\mathrm{Spline}}$ is the bucketing method, the worst-case complexity of the standard sort is $\mathcal{O}(nU(n))$ (where $U(n)$ is a non-decreasing function), and $\alpha = \beta = \gamma = \delta = \lfloor n^{3/4} \rfloor$, then the worst-case complexity of Spline Learned Sort is $\mathcal{O}(nU(n) + n \log \log n)$.*

*Proof.* The proof is identical to that of Theorem 3.5, using Lemma 3.1. $\qquad\square$

**Theorem A.9** (Spline version of Theorem 3.6). *Let $\sigma_1$ and $\sigma_2$ be the lower and upper bounds, respectively, of the probability density $f(x)$ on $\mathcal{D}$, and assume $0 < \sigma_1 \leq \sigma_2 < \infty$. If $\mathcal{M}_{\mathrm{Spline}}$ is the bucketing method, the expected complexity of the standard sort is $\mathcal{O}(n \log n)$, and $\alpha = \beta = \gamma = \delta = \lfloor n^{3/4} \rfloor$, then the expected complexity of Spline Learned Sort is $\mathcal{O}(n \log \log n)$.*

To prove this theorem, we first establish the following lemma.

**Lemma A.10** (Spline version of Lemma A.1). *Define $e, \mathcal{S}_r, \mathcal{T}_r, Y_{jr}, Z_{jr}, Y_r, Z_r$ exactly as in Lemma A.1 ($j = 1, \ldots, \alpha$, $r = 1, \ldots, n$). That is, let $e \in \mathcal{I}^n$ be the sorted version of $\boldsymbol{x}_\mathcal{I} \in \mathcal{I}^n$ and set $\Delta := (x_{\max} - x_{\min})/\beta$ (where $x_{\min}$ and $x_{\max}$ are the minimum and maximum values of $\boldsymbol{x}_\mathcal{I}$, respectively). Also define*

$$\mathcal{S}_r = \{k \mid e_{\max(1, r-\delta/2)} + \Delta < e_k \leq e_r - \Delta\}, \qquad \mathcal{T}_r = \{k \mid e_r + \Delta \leq e_k < e_{\min(r+\delta/2, n)} - \Delta\}, \tag{49}$$

*and*

$$Y_{jr} = \begin{cases} 1 & (j \in \mathcal{S}_r) \\ 0 & (\text{otherwise}) \end{cases}, \qquad Z_{jr} = \begin{cases} 1 & (j \in \mathcal{T}_r) \\ 0 & (\text{otherwise}) \end{cases}, \qquad Y_r = \sum_{j=1}^{\alpha} Y_{jr}, \qquad Z_r = \sum_{j=1}^{\alpha} Z_{jr}. \tag{50}$$

*When using $\mathcal{M}_{\mathrm{Spline}}$, if the size of the bucket to which $e_r$ is assigned is at least $\delta$, then the same condition as in Equation (19) holds; namely,*

$$\left( r \geq \tfrac{\delta}{2} + 1 \ \wedge \ Y_r \leq \left\lfloor \tfrac{\alpha}{\gamma} \right\rfloor \right) \vee \left( r \leq n - \tfrac{\delta}{2} \ \wedge \ Z_r \leq \left\lfloor \tfrac{\alpha}{\gamma} \right\rfloor \right). \tag{51}$$

*Proof.* The proof proceeds exactly as in Lemma A.1, by contraposition. That is, assume $\left( r < \tfrac{\delta}{2} + 1 \ \vee \ Y_r > \left\lfloor \tfrac{\alpha}{\gamma} \right\rfloor \right) \wedge \left( r > n - \tfrac{\delta}{2} \ \vee \ Z_r > \left\lfloor \tfrac{\alpha}{\gamma} \right\rfloor \right)$, and prove that $e_r$ is assigned to a bucket smaller than $\delta$.

First, we show that $e_r$ and $e_{\min(r+\delta/2, n+1)}$ are assigned to different buckets. As in the proof of Lemma A.1, the case $n - \delta/2 < r \leq n$ is immediate; hence we focus on $r \leq n - \delta/2$. The bucket ID of $e_r$ is $\lfloor \tilde{F}(e_r)\gamma \rfloor + 1$, while that of $e_{r+\delta/2}$ is $\lfloor \tilde{F}(e_{r+\delta/2})\gamma \rfloor + 1$. By the definition of the spline-based CDF model and the inequalities $t_{i(e_r)+1} \leq e_r + \Delta$, $t_{i(e_{r+\delta/2})} \geq e_{r+\delta/2} - \Delta$, we obtain

$$\tilde{F}(e_r) \leq F_{\boldsymbol{a}}(t_{i(e_r)+1}) = \tfrac{1}{\alpha} \left| \{j \mid a_j \leq t_{i(e_r)+1}\} \right| \leq \tfrac{1}{\alpha} \left| \{j \mid a_j \leq e_r + \Delta\} \right|, \tag{52}$$

$$\tilde{F}(e_{r+\delta/2}) \geq F_{\boldsymbol{a}}(t_{i(e_{r+\delta/2})}) = \tfrac{1}{\alpha} \left| \{j \mid a_j \leq t_{i(e_{r+\delta/2})}\} \right| \geq \tfrac{1}{\alpha} \left| \{j \mid a_j \leq e_{r+\delta/2} - \Delta\} \right|. \tag{53}$$

Hence, their bucket IDs satisfy

$$\lfloor \tilde{F}(e_r)\gamma \rfloor + 1 \leq \tfrac{\gamma}{\alpha} |\{j \mid a_j \leq e_r + \Delta\}| + 1, \tag{54}$$

$$\lfloor \tilde{F}(e_{r+\delta/2})\gamma \rfloor + 1 > \tfrac{\gamma}{\alpha} |\{j \mid a_j \leq e_{r+\delta/2} - \Delta\}|, \tag{55}$$

which are identical to Equations (20) and (21). Thus,

$$\left( \lfloor \tilde{F}(e_{r+\delta/2})\gamma \rfloor + 1 \right) - \left( \lfloor \tilde{F}(e_r)\gamma \rfloor + 1 \right) > 0, \tag{56}$$

and hence $e_r$ and $e_{r+\delta/2}$ are assigned to different buckets.

By a symmetric argument, $e_{\max(0, r-\delta/2)}$ and $e_r$ are also assigned to different buckets. Therefore, the bucket containing $e_r$ has size at most $\delta - 1$, completing the contraposition. $\qquad \square$

From Lemma A.10, we obtain the following theorem.

**Theorem A.11** (Spline version of Lemma 3.4). *Let $\sigma_1$ and $\sigma_2$ be the lower and upper bounds of the probability density $f(x)$ on $\mathcal{D}$, and assume $0 < \sigma_1 \leq \sigma_2 < \infty$ (i.e., $\sigma_1 \leq f(x) \leq \sigma_2$ for all $x \in \mathcal{D}$). Then, in model-based bucketing of $\boldsymbol{x}_\mathcal{I} \in \mathcal{I}^n$ into $\{\boldsymbol{c}_j\}_{j=1}^{\gamma+1}$ using $\mathcal{M}_{\mathrm{Spline}}$, the following holds for any interval $\mathcal{I} \subseteq \mathcal{D}$:*

$$K := \frac{\gamma \delta}{2n} - \frac{2\sigma_2 \gamma}{\sigma_1 \beta} \geq 1 \ \Rightarrow \ \Pr[\exists j, |\boldsymbol{c}_j| > \delta] \leq \frac{2n}{\delta} \exp\left\{ -\frac{\alpha K}{2\gamma} \left(1 - \tfrac{1}{K}\right)^2 \right\}. \tag{57}$$

*Proof.* The proof is identical to that of Lemma 3.4, using Lemma A.10. $\qquad \square$

Table 1: Statistics of real-world datasets used in our experiments.

| Dataset | Size | # Unique Values | % of Duplicates |
|---|---|---|---|
| Chicago [Start] (Chicago, 2021) | 39,588,772 | 216,610 | 99.45 |
| Chicago [Tot] (Chicago, 2021) | 39,199,154 | 6,887 | 99.98 |
| NYC [Pickup] (nyc, 2020) | 100,000,000 | 26,666,741 | 73.33 |
| NYC [Dist] (nyc, 2020) | 200,107,656 | 2,060 | 100.00 |
| NYC [Tot] (nyc, 2020) | 199,964,459 | 7,428 | 100.00 |
| SOF [Humidity] (Mavrodiev, 2019) | 96,318,228 | 8,128 | 99.99 |
| SOF [Pressure] (Mavrodiev, 2019) | 96,317,180 | 862,502 | 99.10 |
| SOF [Temperature] (Mavrodiev, 2019) | 96,432,856 | 5,638 | 99.99 |
| Wiki (Marcus et al., 2020) | 200,000,000 | 90,437,011 | 54.78 |
| OSM (Marcus et al., 2020) | 800,000,000 | 799,469,195 | 0.07 |
| Books (Marcus et al., 2020) | 800,000,000 | 799,994,961 | 0.00 |
| Face (Marcus et al., 2020) | 199,998,000 | 199,998,000 | 0.00 |
| Stocks [Volume] (Onyshchak, 2020) | 27,596,686 | 325,654 | 98.82 |
| Stocks [Open] (Onyshchak, 2020) | 27,596,686 | 830,758 | 96.99 |
| Stocks [Date] (Onyshchak, 2020) | 27,596,686 | 14,646 | 99.95 |
| Stocks [Low] (Onyshchak, 2020) | 27,596,686 | 863,701 | 96.87 |

Finally, using Theorem A.11, we can prove Theorem A.9.

*Proof of Theorem A.9.* The proof follows exactly the same steps as Theorem 3.6, applying Theorem A.11 to Lemma 3.3. □

## B  Real Dataset Details

In the main text, we provided only a concise list of the real datasets used in our experiments. Here we describe each dataset in more detail.

- **Chicago [Start, Tot]**: The Chicago Taxi Trips dataset includes taxi trips reported to the City of Chicago in its role as a regulatory agency over the last six years. The data to be sorted includes trip starting timestamps and total fare amounts.

- **NYC [Pickup, Dist, Tot]**: The New York City yellow taxi trip dataset includes trip pickup datetimes, trip distances, and total fare amounts.

- **SOF [Humidity, Pressure, Temperature]**: The Sofia dataset contains time-series air quality metrics (humidity, pressure, and temperature) measured at 1-minute intervals from outdoor sensors in Sofia, Bulgaria.

- **Wiki**: The Wikipedia dataset contains article edit timestamps (Marcus et al., 2020).

- **OSM**: Uniformly sampled OpenStreetMap locations represented as Google S2 CellIds (Marcus et al., 2020).

- **Books**: Book sale popularity data from Amazon (Marcus et al., 2020).

- **Face**: The FB dataset contains an upsampled set of Facebook user IDs obtained via random walks on the FB social graph (Marcus et al., 2020). As in (Kristo et al., 2021a; Ferragina & Odorisio, 2025), the outliers greater than the 0.99999 quantile are discarded.

- **Stocks [Volume, Open, Date, Low]**: The Stocks dataset contains historical daily opening, low prices, trading volumes, and dates for all NASDAQ tickers (stocks and ETFs), retrieved via the yfinance Python package up to April 1, 2020.

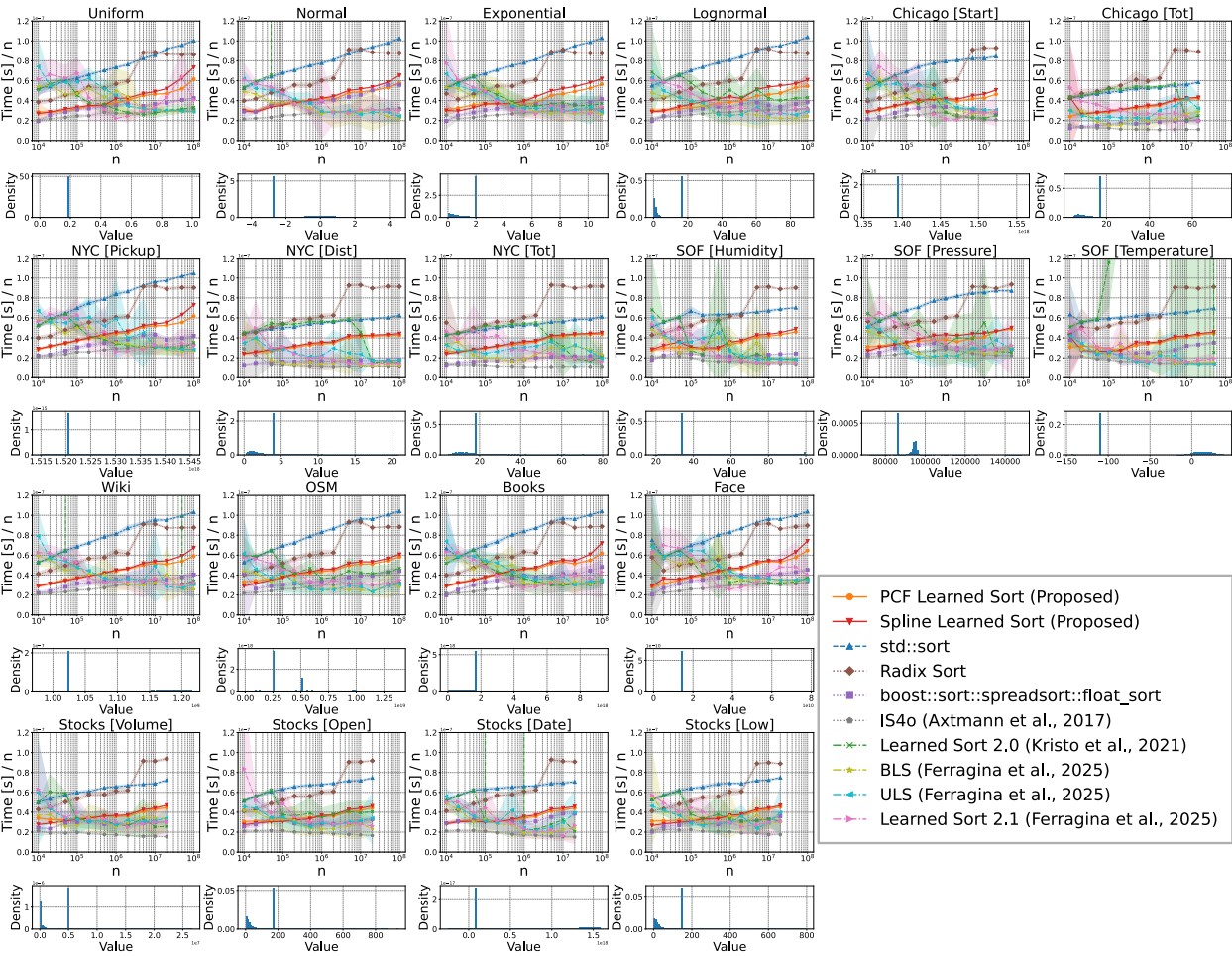

Figure 5: Time to sort the array in adversarial environments. Below each graph is a histogram that visualizes the distribution of each dataset. The standard deviation of the 10 measurements is represented by the shaded area.

Furthermore, the detailed statistics of these datasets, including their sizes, the number of unique values, and the percentage of duplicates, are summarized in Table 1. The percentage of duplicates is computed as $100 \times \left(1 - \frac{|\,\text{unique\_values}\,|}{n}\right)$, where $n$ is the total number of elements and `unique_values` is the set of distinct elements. In particular, for several datasets (Chicago [Start, Tot], NYC [Dist, Tot], SOF [Humidity, Pressure, Temperature], and Stocks [Volume, Open, Date, Low]), the number of unique values is extremely small, accounting for less than $3.2\%$ of the total elements.

## C   Experiments in Adversarial Environments

To evaluate the robustness of our Learned Sort, we conducted experiments under adversarially constructed inputs that explicitly violate the assumptions of Lemma 3.3. Specifically, given an original dataset of size $n$, we injected $n$ duplicate elements of a randomly chosen value inside the range $[x_{\min}, x_{\max}]$, where $x_{\min}$ and $x_{\max}$ is the minimun and maximum value of the input array $\boldsymbol{x}$. As a result, the array length doubled to $2n$, and the constructed distribution contained a point mass of probability one-half, leading to a huge probability density at that value. This setting intentionally breaks the assumption $\sigma_2 < \infty$ required in Lemma 3.3.

Figure 5 reports the sorting time on such adversarial datasets. The results show that the expected bound of $\mathcal{O}(n \log \log n)$ for our PCF Learend Sort no longer holds once the assumptions are violated, and the performance of PCF Learned Sort occasionally degrades to the level of `std::sort`. However, we consistently observed that PCF Learned Sort never exceeded the worst-case complexity of $\mathcal{O}(n \log n)$, and thus its runtime remained comparable to `std::sort`.

On the other hand, we found that Learned Sort 2.0 (Kristo et al., 2021b), which has $\mathcal{O}(n^2)$ worst-case complexity, sometimes exhibited severe slowdowns. For example, on data of size $n = 10^5$ sampled from a Normal distribution with injected duplicates, the sorting time of Learned Sort 2.0 reached up to 10.89 seconds, whereas PCF Learned Sort required at most 0.021 seconds.

We also found that BLS, ULS, and Learned Sort 2.1 (Ferragina & Odorisio, 2025), which also have $\mathcal{O}(n^2)$ worst-case complexity, sometimes exhibited substantial performance degradation. For example, on data of size $n = 10^6$ sampled from a Normal distribution with injected point masses, Learned Sort 2.1 took up to 0.154 seconds, compared with 0.042 seconds for PCF Learned Sort and 0.078 seconds for `std::sort`.

These results highlight the fragility of sorting algorithms without strong worst-case complexity guarantees, a limitation shared by many existing learned sorting algorithms. They underscore the importance of developing learned sorting algorithms—such as our PCF Learned Sort—that combine practical efficiency with rigorous worst-case complexity guarantees.

## D    Runtime Profiling

To better understand the practical behavior of PCF Learned Sort, we profiled the runtime of its major components: (i) model training, (ii) bucketing, (iii) standard sort applied to buckets that are too small ($< \tau$), and (iv) standard sort applied to buckets that are too large ($\geq \delta$). The results are shown in Figure 6.

For small input sizes $n$, the majority of the runtime is typically spent on applying the standard sort to buckets smaller than $\tau$. As $n$ increases, the relative cost shifts, and most of the runtime is instead consumed by the bucketing stage. Model training remains consistently lightweight, comparable to or faster than bucketing, and never exceeds roughly one quarter of the total runtime. Thus, training does not become the dominant factor in practice.

Figure 7 shows the results for adversarially clustered inputs (as described in Appendix C). Under the adversarial setting, the bucketing algorithm more frequently produces large buckets, which are then handled by standard sort. In these cases, a larger fraction of the runtime is attributed to this fallback mechanism.

Overall, these analyses clarify which components of PCF Learned Sort dominate the runtime under different conditions. They also highlight that the algorithm adapts gracefully: training and bucketing scale well, while the worst-case fallback ensures robustness without catastrophic slowdowns.

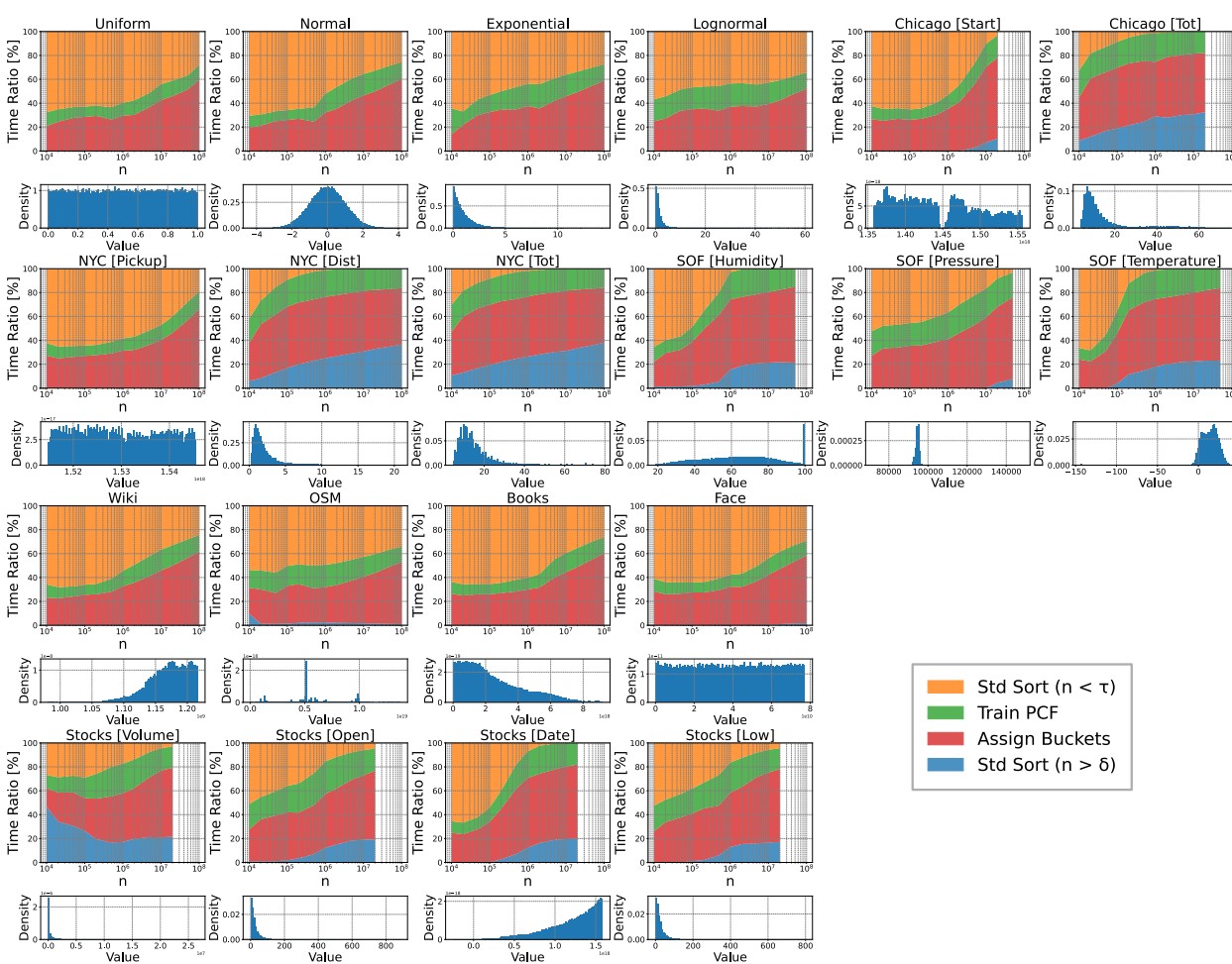

Figure 6: Breakdown of runtime components in PCF Learned Sort. The total runtime is decomposed into (i) model training, (ii) bucketing, (iii) standard sort applied to buckets smaller than $\tau$, and (iv) standard sort applied to buckets larger than $\delta$.

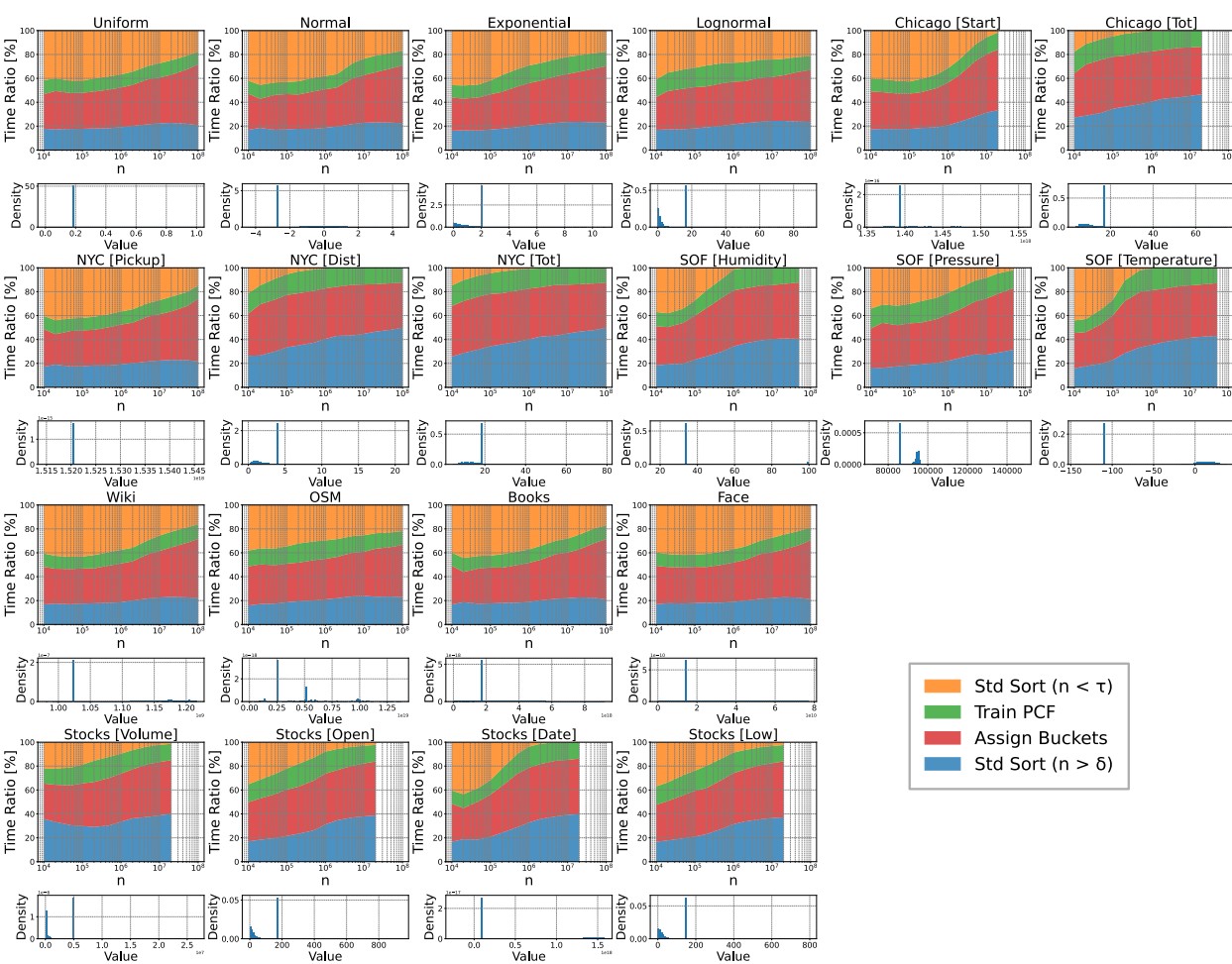

Figure 7: Breakdown of runtime components in PCF Learned Sort in adversarial environments. The total runtime is decomposed into (i) model training, (ii) bucketing, (iii) standard sort applied to buckets smaller than $\tau$, and (iv) standard sort applied to buckets larger than $\delta$.

