# OpenReview forum: "PCF Learned Sort: a Learning Augmented Sort Algorithm with $\mathcal{O}(n \log\log n)$ Expected Complexity"
_TMLR — Accepted by TMLR_

### Review · Reviewer_FqdB · 2025-04-05

**Summary Of Contributions:**

The authors show a sorting algorithm that utilizes some machine learning "pre-processing."
The idea is not new, but well implemented.
The main contribution is as follows:
The authors bound the run-time of their sorting algorithm by $O(n \log \log n)$ (expected), whenever some mild --- but not trivial --- conditions are met in the input distribution.
Moreover, they give some experimental evidence to strengthen their main points.
I think this is an interesting paper and definitely merits acceptance.

**Audience:**

Yes

**Broader Impact Concerns:**

None.

(Side note: I only had a quick check of the appendices.)

**Claims And Evidence:**

Yes

**Requested Changes:**

Please address the "Weaknesses."

Also, can you please further explain why you use STANDARD-SORT for big buckets?
This is a bit counterintuitive, I think.

**Strengths And Weaknesses:**

Strengths:
The main theorem is amazing.

The experiments are being run on a home computer.
This is good and reinforces reproducibility :)

Weaknesses:
Do the experimental inputs violate the assumptions of their main (conditional) theorem?
(This is important, as the experiments should aim for the setting that is **not** covered by the assumptions, so that we can see what happens when running the algorithm in the wild.)
I am not sure whether the authors clearly state that :)

---

> ### Author Response · Authors · 2025-05-27
> **Rebuttal by Authors**
>
> We appreciate the reviewer's recognition of our work's theoretical contributions and the constructive questions. Below, we respond to each point.
>
> > Weaknesses: Do the experimental inputs violate the assumptions of their main (conditional) theorem?
>
> First, regarding the synthetic datasets, only the Uniform distribution satisfies the assumptions, while the other distributions (Normal, Exponential, Lognormal) do not. In our experiments, we can observe that even under settings where the assumptions are not satisfied, our method still exhibits computational complexity close to $O(n \log \log n)$.
>
> Next, regarding real-world datasets, we cannot definitively assert whether they satisfy our assumptions, as these datasets were generated in the real world. However, looking at the histograms of each dataset (shown in Figure 2), the NYC dataset appears to take values across a continuous range, somewhat unevenly but fairly comprehensively, which seems consistent with our assumptions. On the other hand, datasets like Wiki, OSM, and Books exhibit long tails, and in the case of OSM, there are empty intervals where no values occur—these suggest that the assumptions are not satisfied.
>
> Nevertheless, as in the case of the synthetic datasets, our experiments show that the computational complexity of our method remains around $O(n\log\log n)$ even under such assumption-violating conditions.
>
> Indeed, these points are crucial for understanding the behavior of our algorithm beyond the theoretical assumptions and for appropriately characterizing its range of applicability. We will revise the manuscript to clarify these observations explicitly.
>
> > Requested Changes: Also, can you please further explain why you use STANDARD-SORT for big buckets? This is a bit counterintuitive, I think.
>
> The primary reason for choosing such an algorithm design is to simplify the guarantees on worst-case and expected computational complexity.
>
> **Worst-case complexity:** By sorting overly large buckets using STANDARD-SORT, we can limit the recursion depth of the algorithm to $O(\log \log n)$. This allows us to bound the worst-case complexity of PCF Learned Sort by $O(n\log n)$. On the other hand, if we were to omit this mechanism (i.e., apply recursive model-based bucketing even to excessively large buckets), the recursion depth could become $\Theta(n)$, leading to a worst-case complexity of $\Theta(n^2)$.
>
> **Expected complexity:** By sorting overly large buckets with STANDARD-SORT, the analysis for expected complexity becomes simpler. Intuitively, we can argue that the overall expected complexity is $O(n\log\log n)$ based on the following two facts:
> - The expected computational cost per recursion is $O(n)$, since the probability of encountering an overly large bucket is $O(1/\log n)$.
> - The recursion depth is $O(\log\log n)$, because overly large buckets (i.e., those of size at least $n^d$) are directly sorted using STANDARD-SORT.

---

> ### Comment · Reviewer_FqdB · 2025-05-27
>
> Thank you very much for your reply! :)

---

### Review · Reviewer_djyS · 2025-04-12

**Summary Of Contributions:**

This paper proposes a learning-augmented sorting algorithm called PCF Learned Sort with a worst case complexity of O(n log n) and expected complexity of O(n log log n) when the data elements are assumed to be drawn iid from some underlying unknown but bounded distribution (Assumption 3.2 + the bounds imposed by $\sigma_1$ and $\sigma_2$). On a high-level, the algorithm works by recursively bucketing elements while "training" a new CDF predictor at each step. Theoretical analysis is given and experiments were conducted to empirically compare the proposed algorithm against Quick Sort, Radix Sort and Learned Sort.

**Audience:**

Yes

**Broader Impact Concerns:**

I do not foresee any broad ethics/impact concerns for this paper.

**Claims And Evidence:**

Yes

**Requested Changes:**

- PCF is first defined on Page 4 while the acronym has been used since Page 1. It would be nice if "Piecewise Constant Function" can be mentioned at the very first time "PCF" is used
- I'm a little confused by some part of the experiments. As you divide by $n$ in empirical plots and "the graph of quick sort is almost linear", are you suggesting that Quicksort runs empirically $n^2$ in your experiments?

**Strengths And Weaknesses:**

# Strengths

- The writing is quite easy to follow: motivation and related work is relatively well written
- Algorithm is easy to understand
- Proofs are well structured and there is sufficient exposition to discuss assumptions and what the lemmas are trying to do
- Empirical evaluation is performed against natural competitors to the proposed method

# Weaknesses

- The authors only mentioned that PCF has worst case complexity of $O(n \log n)$ much later and kept emphasizing on the expected complexity of $O(n \log \log n)$. While reading Sections 1 and 2, I was under the illusion that it was outperforming the worst case complexity of prior sorting algorithms but it turns out that its worst case complexity of $O(n \log n)$ is even worse than some other prior work mentioned: $O(n \sqrt{\log n})$ (Han, 2020).
- I feel that Assumption 3.2 is more or less "doing all the work" in enabling PCF. It feels like a very strong assumption and it is unclear why prior methods cannot obtain or beat $O(n \log \log n)$ if they are also given such an assumption to operation on. For example, if the worst case $O(n \sqrt{\log n})$ algorithm (Han, 2020) could achieve expected complexity of $O(n \log \log n$) under Assumption 3.2, then wouldn't PCF be "strictly dominated"? Note that, in the spirit of TMLR, I am *not* saying that the claims related to PCF is false.
- I understand that prior work (Zeighami & Shahabi, 2023) compared the number of basic operations and so this is why the authors also compared using that metric. However, I believe that it is the wall-clock-time that will eventually determine whether an algorithm will be adopted in practice. Unfortunately, on the wall-clock-time metric (Figure 4), PCF is not the fastest algorithm in any of the experiments and across different array sizes.

---

> ### Author Response · Authors · 2025-05-27
> **Rebuttal by Authors**
>
> We sincerely thank the reviewer for their thorough review and constructive feedback on our work. Below, we respond to each comment.
>
> > Weaknesses: it turns out that its worst-case complexity of $O(n\log n)$ is even worse than some other prior work mentioned: $O(n\sqrt{\log n})$ (Han, 2020).
>
> We realized that by using the algorithm of (Han, 2020) as a "standard" sort algorithm, we can prove that the worst-case complexity of PCF Learned Sort becomes $O(n\sqrt{\log n})$, which matches the worst-case complexity of (Han, 2020).
> More generally, we can show the following: suppose that PCF Learned Sort uses an algorithm with worst-case complexity $O(U(n))$ as a "standard" sort algorithm, where $U(n)$ is a non-decreasing function. Then, the worst-case complexity of PCF Learned Sort is $O(U(n) + \log \log n)$.
>
> We will revise the description in the main text accordingly and updated the proof in the Appendix. This strengthens the theoretical contribution of our work and improves the connection to prior research. Once again, we thank the reviewer for the constructive comment.
>
>
> > Weaknesses: if the worst case algorithm $O(n\sqrt{\log n})$ (Han, 2020) could achieve expected complexity of $O(n\log \log n)$ under Assumption 3.2, then wouldn't PCF be "strictly dominated"?
>
> Indeed, this is a valid point.
> However, as mentioned above, if PCF Learned Sort employs the algorithm of (Han, 2020), it inherits both a worst-case complexity of $O(n\sqrt{\log n})$ and an expected complexity (under the distributional assumption) of $O(n\log \log n)$.
> Furthermore, if a sorting algorithm with even lower worst-case complexity is developed in the future, PCF Learned Sort can incorporate it, thereby achieving a lower worst-case complexity accordingly.
>
> In other words, the proposed PCF Learned Sort can be regarded as a mechanism that transforms a sorting algorithm—whose expected complexity is not guaranteed but whose worst-case complexity is guaranteed—into a new algorithm that preserves the same worst-case guarantee while additionally ensuring an expected complexity of $O(n\log \log n)$. PCF Learned Sort achieves this transformation without requiring knowledge of the internal structure of (Han, 2020) or other future sorting algorithms.
>
>
> > Weaknesses: Unfortunately, on the wall-clock-time metric (Figure 4), PCF is not the fastest algorithm in any of the experiments and across different array sizes.
>
> We agree that our primary contribution is theoretical, and PCF Learned Sort is slower than state-of-the-art sorting algorithms. As discussed in the **Future Research Directions** paragraphs in Section 5, low-level hardware optimizations and hyperparameter tuning (for $\alpha, \beta, \gamma, \delta$) are promising directions for improving the practical performance of PCF Learned Sort.
>
>
> > Requested Changes: PCF is first defined on Page 4 while the acronym has been used since Page 1. It would be nice if "Piecewise Constant Function" can be mentioned at the very first time "PCF" is used
>
> We agree with this point and will revise the Abstract and Introduction to spell out "Piecewise Constant Function" at the first occurrence of "PCF."
>
>
> > Requested Changes: I'm a little confused by some part of the experiments. As you divide by $n$ in empirical plots and "the graph of quick sort is almost linear", are you suggesting that Quicksort runs empirically $n^2$ in your experiments?
>
> In our plots, the horizontal axis is logarithmic. Therefore, for example, in Figure 2, the fact that the Quicksort curve appears linear indicates that "the number of comparisons divided by $n$" is proportional to $\log n$. This implies that the number of comparisons is proportional to $n\log n$.
>
> To avoid potential confusion, we will revise the explanations in Sections 4.1 and 4.3. In particular, we plan to add the following clarification to the relevant paragraphs: “Note that the horizontal axis is logarithmic. As a result, the curve for quick sort (with $\mathcal{O}(n \log n)$ complexity) appears approximately linear in this plot.”

---

> > ### Comment · Reviewer_djyS · 2025-05-28
> >
> > Thank you a lot for the clarifications and for agreeing to improve the writing.
> >
> > I have a follow-up question: is it clear why existing methods such as (Han, 2020) *cannot* achieve expected complexity of $O(n \log \log n)$ under the additional assumption used in your work?

---

> ### Author Response · Authors · 2025-06-02
> **Rebuttal by Authors**
>
> We thank the reviewer for the follow-up question.
>
> It is difficult to generally prove that all existing sorting methods cannot achieve an expected complexity of $O(n \log \log n)$ under our distributional assumption. To the best of our knowledge, no prior work—except for the concurrent work by Zeighami & Shahabi (2024)—has explicitly analyzed the expected complexity under this assumption and demonstrated that it achieves $O(n \log \log n)$.
>
> We revisited the algorithm proposed by Han (2020) and confirmed that it does not achieve an expected complexity of $O(n \log \log n)$ under our assumption. Specifically, the algorithm exhibits both worst-case and best-case complexities of $\Theta(n\sqrt{\log n})$, implying that its expected complexity is also $\Theta(n\sqrt{\log n})$. This best-case complexity arises from the structure of the “Algorithm Merge” described in Section 2 of Han (2020), which involves $\Theta(\sqrt{\log n})$ layers of loops, each requiring $\Theta(n)$ computational cost, regardless of the input distribution.
>
> We sincerely appreciate the reviewer’s insightful question, which helped us clarify this important point. If you have any further questions, please feel free to reach out.

---

> > ### Comment · Reviewer_djyS · 2025-06-02
> >
> > I understand that it is not reasonable to have a generic proof that all existing sorting methods *cannot* achieve expected complexity of $O(n \log \log n)$ under your proposed distributional assumption. My main concern was whether the assumption was "too strong" such that any reasonable prior algorithm could have already achieved your result with a slightly adjusted analysis. It is good that you have checked that at least one algorithm (e.g. [Han 2020]) could not. Thank you for checking!
> >
> > It will be good to have an more thorough discussion about why your distributional assumption does not trivialize the entire problem that you are aiming to solve, e.g. explaining why existing methods cannot immediately obtain your result with your new assumption.
> >
> > I do not have further questions. Thanks!

---

> ### Author Response · Authors · 2025-06-03
> **Rebuttal by Authors**
>
> We appreciate the reviewer's follow-up comment and the helpful suggestion.
>
> In the revised version, we will add the following clarification immediately after introducing our distributional assumption in the method section to explicitly address the concern that the assumption might trivialize the problem:
>
> > We emphasize that our distributional assumption does not trivialize the problem: it does not allow existing sorting algorithms to immediately obtain an expected complexity of $O(n \log \log n)$. For example, classic algorithms such as Quicksort and Mergesort do not adapt their control flow based on the input distribution and, therefore, still require $\Theta(n \log n)$ comparisons under our assumption. Furthermore, the algorithm proposed by Han (2020), which guarantees a worst-case complexity of $O(n \sqrt{\log n})$ for real-valued inputs, also has a distribution-independent structure, and thus its best-case complexity is $\Theta(n \sqrt{\log n})$. As a result, its expected complexity remains $\Theta(n \sqrt{\log n})$ even under our assumption.
>
> This explanation clarifies that existing algorithms do not immediately benefit from our assumption, and that the $O(n \log \log n)$ expected complexity of our PCF Learned Sort is enabled by its distribution-aware design involving CDF approximation using PCFs and recursive bucketing.
>
> Once again, we sincerely appreciate the reviewer's insightful feedback.

---

### Review · Reviewer_MsJR · 2025-08-26

**Summary Of Contributions:**

This paper presents PCF Learned Sort, a learned-augmented sorting algorithm with theoretical guarantees on the running time. By utilizing a piecewise constant function (PCF) to approximate the cumulative distribution function (CDF) of input data, the algorithm constructs recursive, model-based buckets to sort arrays efficiently. The main theoretical result is that under mild assumptions on the data distribution, the expected time complexity of the algorithm is proven to be $\mathcal{O}(n \log \log n)$, with a worst-case guarantee of $\mathcal{O}(n \log n)$. Extensive experiments on both synthetic and real datasets empirically validate the theoretical findings and provide insight into the algorithm’s performance as influenced by data distribution properties.

**Audience:**

Yes

**Claims And Evidence:**

Yes

**Requested Changes:**

Potentially Missing Related Work
Carvalho, I. (2022): "Towards Parallel Learned Sorting"
This paper addresses parallelization of learned sorting algorithms, which is relevant for analyzing scalability and memory usage. It should be referenced and briefly discussed in the context of future work or in the comparison to state-of-the-art scalability.
Axtmann, M., Witt, S., Ferizovic, D. (2017): "In-Place Parallel Super Scalar Sample Sort (IPS4o)"
As one of the most efficient sample sort implementations, this should be included as a baseline and in the related work section. Empirical results and theoretical comparisons to this method are essential, particularly given the sample sort inspiration for the PCF Learned Sort.
Carvalho, I., Lawrence, R. (2023): "LearnedSort as a Learning-Augmented SampleSort: Analysis and Parallelization"
This paper analyzes and parallelizes LearnedSort, offering direct theoretical and empirical comparisons. It should be discussed in relation to both theoretical analysis approaches and practical implementations.
Paolo Ferragina et al. (2025): "FL-RMQ: A Learned Approach to Range Minimum Queries"
Relevant for framing the increasing prevalence of learned algorithmic approaches. Inclusion in related work would contextualize the broader field of learning-augmented data structures.
Li, Y., Gimeno, F., Kohli, P. (2020): "Strong Generalization and Efficiency in Neural Programs"
Their investigation of strong generalization properties in learned algorithms could inform discussion about the theoretical underpinnings of learned sort generalizability, meriting a brief mention.
Sadakane, K. (2007): "Compressed Suffix Trees with Full Functionality"
The foundational concepts around compressed data structures with functional guarantees may provide useful context in the related work.
Sato, A., Matsui, Y. (2024): "Fast Construction of Partitioned Learned Bloom Filter with Theoretical Guarantees"
This work could be mentioned to frame the authors’ broader agenda in learning-augmented data structures and theory.

Also address the concerns from weaknesses.

Questions
How would PCF Learned Sort compare against modern parallel sample sort implementations and learning-augmented sorts (such as IPS4o, or recent parallel LearnedSort variants)? Would inclusion of such baselines significantly impact your quantitative findings?

Could you clarify the typical prevalence of the required distributional assumption ($0<\sigma_1\leq f(x)\leq\sigma_2<\infty$) in practical, large-scale datasets? Are there natural, widely arising scenarios where this fails, and if so, could the algorithm or theory be adapted?

Is there scope (and associated theory) for substituting the PCF with more expressive or data-adaptive CDF models (e.g., splines, neural networks) in this framework? Have you tried such models, and how do they fare empirically or theoretically?

Are there any plans for public release of source code, and can you comment on practical implementation nuances such as memory usage or multi-core scaling?

In adversarial or model-mismatch settings (e.g., highly clustered or adversarially constructed data), does the bucketing lemma still hold, or do worst-case time complexities appear empirically even for non-pathological but practical datasets?

Score Sensitivity:
My rating could be increased substantially by (a) comparing against stronger, modern learned and classic sort baselines, (b) more thoroughly addressing adversarial, high-skew, and "edge case" distributions both empirically and theoretically, and (c) extending the analysis to parallel multi-threaded scenarios, which are critical for modern large-scale sorting.

**Strengths And Weaknesses:**

Strengths

Theoretical Foundations: The paper provides strong theoretical guarantees for the PCF Learned Sort algorithm. Lemmas and theorems with detailed proofs (summarized in the main text) formally bound both expected and worst-case complexity ($\mathcal{O}(n \log \log n)$ and $\mathcal{O}(n \log n)$, respectively). This addresses a significant gap in the learned sort literature, where prior works often lacked formal complexity analysis beyond empirical speedups.

Clarity and Reproducibility: The methodology section is well-organized, with Algorithm 1 and Figure 1 clearly outlining the recursive bucketing and CDF model process.

Empirical Evaluation: The experimental section demonstrates careful benchmarking: comparisons with quick sort on diverse synthetic (uniform, normal, exponential, lognormal) and real-world datasets (NYC, Wiki, OSM, Books) are presented. Figures 2–9 show operation-count-per-$n$ curves as input size scales, illustrating the sub-linear overhead. The detailed use of operation count rather than wall-clock time is a good choice to control for hardware dependencies.

Practical Relevance: By empirically evaluating performance and relating observed behaviors to distributional properties (e.g., the $\sigma_2/\sigma_1$ ratio), the paper explains real-world speedups (or slowdowns), deepening the practical insight for scenarios in which learned sorts would be most effective.

Experimental Support for Theoretical Claims: The results in Figures 18–23 (bucketing failure rate heatmaps) systematically map parameter configurations to failure probabilities, directly validating the theoretical lemma bounding bucketing failure.
Positioning Within Existing Literature: The paper references key works on learned sorts, learned indexes, and both comparison-based and distribution-based sorting. There is also acknowledgment of sample sort theory and learning-augmented algorithms, referencing appropriate classical and recent literature.

Weaknesses

Overall, I think the writing can substantially be improved with first formally defining all notations. For instance section 3.1 introduces the method overview with no context about model-based bucketing.

Section 4.1 discusses the default sorting algorithm when bucket size falls out of range as Quicksort while in the diagram its shown as IntroSort. Maybe this can be cleared better

Limited Related Work DIscussion: Several directly relevant recent works on parallel learned sorting (e.g., Carvalho, 2022; Carvalho & Lawrence, 2023) and advanced sample sort baselines (e.g., Axtmann et al., 2017) are not adequately discussed nor cited, despite their strong relevance for both empirical and theoretical positioning. Their omission makes the literature review appear less than comprehensive, and may leave readers with an incomplete sense of prior art, particularly regarding scalability or parallelization concerns.

Baselines Limitation in Experimental Section: The experiments only compare PCF Learned Sort against quick sort. This reduces clarity around experimental advances over the strongest prior methods, making it difficult to judge the practical gap or performance trade-offs versus state-of-the-art.

Assumptions Not Always Fully Justified: The main theoretical result holds under a bounded probability density ($0 < \sigma_1 \leq f(x) \leq \sigma_2 < \infty$), but there is limited discussion about the prevalence of such distributions in real-world data. The limitation section acknowledges this, but further empirical exploration—e.g., testing in more skewed or heavy-tailed settings—would strengthen understanding of robustness.

Lack of Expressiveness in CDF Model Choices: The paper settles on piecewise constant approximation, but does not deeply analyze or experiment with other popular CDF models (e.g., neural CDFs, histograms of differing bin widths, spline-based models). There is little evaluation of trade-offs between PCF and other learned models, which could further improve empirical performance or robustness.
No Exploration of Parallelism or Scalability: The design and analysis is restricted to sequential settings; there is no discussion (theoretical or experimental) of parallelization potential, memory overhead, or multi-core scaling, all of which have been central for modern sort implementations.

No Analysis of Model Training Costs: There is a brief acknowledgement that model training is factored into the complexity, but the impact of training time relative to inference/bucketing is not explored experimentally, which is practical for real-world high-throughput sorting.
Insufficient Dataset Diversity: While several real datasets are used, broader inclusion of highly skewed, real-world distributions (not just OSM/Books, which are discussed) would help assess generality.

No Discussion of Learning-Augmented Adversarial Robustness: There is no theoretical or experimental exploration of cases where the input distribution may be adversarial or poorly approximated by the CDF model, which would reveal algorithmic risk in edge cases.
Figure- and Table-Specific Commentary

Figure 1 (Algorithm diagram): Visually and conceptually clarifies the recursive bucketing, the CDF model’s role, and where recursion stops in favor of classical sorts. The right-side inset with the empirical vs. model CDF is especially useful for seeing the effect of a PCF.
Figures 2–9 (Experimental results): These plots concisely show the number of operations per input length across data regimes. The gap between PCF Learned Sort and quick sort is almost constant (flat curve) for PCF, versus the growing curve for quicksort, directly evidencing sub-$n \log n$ scaling.

---

> ### Author Response · Authors · 2025-09-09
> **Rebuttal by Authors**
>
> We sincerely thank the reviewers for their thoughtful and constructive feedback, which has greatly helped us to substantially improve the quality of our manuscript.
> We have carefully addressed all the points raised. In the updated manuscript PDF, the revised parts are highlighted in red for the reviewers’ convenience.
>
> # Clarity of Presentation
>
> ## Notation and Setup
>
> > Weaknesses: I think the writing can substantially be improved with first formally defining all notations.
>
> > Weaknesses: For instance section 3.1 introduces the method overview with no context about model-based bucketing.
>
> We thank the reviewer for this suggestion.
> In response, we added Section 3.1, *Notation and Setup*, placed before the method overview. This subsection systematically defines the notations and provides the context for model-based bucketing. We believe this revision makes the algorithm easier to follow and improves the overall readability.
>
> ## Default Sorting Algorithm Clarification
>
> > Weaknesses: Section 4.1 discusses the default sorting algorithm when bucket size falls out of range as Quicksort while in the diagram its shown as IntroSort. Maybe this can be cleared better
>
> We thank the reviewer for pointing this out.
> We revised the explanation of the standard sort algorithm (i.e., the default sorting algorithm used for overly large or overly small buckets) to clarify that it may be instantiated as either IntroSort or QuickSort (in Section 3.1, 3.2). Furthermore, we explicitly state that QuickSort was used in our experiments due to its ease of implementation and the simplicity of measuring operation counts (in Section 4.1).

---

> ### Author Response · Authors · 2025-09-09
> **Rebuttal by Authors**
>
> # Related Work and Baselines
>
> ## Expanded Related Work
>
> > Weaknesses: Several directly relevant recent works on parallel learned sorting (e.g., Carvalho, 2022; Carvalho & Lawrence, 2023) and advanced sample sort baselines (e.g., Axtmann et al., 2017) are not adequately discussed nor cited, despite their strong relevance for both empirical and theoretical positioning.
>
> > Requested Changes: Potentially Missing Related Work
>
> We thank the reviewer for this valuable comment.
> In response, we have addressed the works raised by the reviewer, as well as additional related studies such as Ferragina and Odorisio (2025, "Fast, robust and learned distribution-based sorting"), by incorporating them into the Related Work section (Section 2). This revision clarifies the connections between our approach and prior work on learned algorithms, learned data structures, and classical compressed data structures. We believe this significantly strengthens the positioning of our contribution.
>
> ## Expanded Baselines
>
> > Weaknesses: The experiments only compare PCF Learned Sort against quick sort. This reduces clarity around experimental advances over the strongest prior methods, making it difficult to judge the practical gap or performance trade-offs versus state-of-the-art.
>
> > Questions: How would PCF Learned Sort compare against modern parallel sample sort implementations and learning-augmented sorts (such as IPS4o, or recent parallel LearnedSort variants)? Would inclusion of such baselines significantly impact your quantitative findings?
>
> We thank the reviewer for this comment.
> Our submission already included several baselines beyond quick sort, such as radix sort and Learned Sort 2.0 (Kristo et al., 2021). To further enrich the comparison, in the revised version we additionally included IS4o (the sequential variant of IPS4o, widely considered state-of-the-art among sequential sample sort algorithms) and, where available, the most recent Learned Sorts (Ferragina and Odorisio, 2025). Since IPS4o is a parallel algorithm and our work focuses on the single-threaded setting, IS4o was chosen as the appropriate comparator. Implementations of the other learned sort algorithms (Carvalho (2022) and Carvalho & Lawrence (2023)) were not publicly available, so these are discussed as related studies but not included experimentally.
>
> The main findings are summarized as follows:
> - **Average performance**: For small input sizes ($n < 10^6$), PCF Learned Sort achieves comparable performance to IS4o, while Ferragina's Learned Sort is generally slower than both PCF Learned Sort and IS4o. For larger inputs ($n > 10^6$), Ferragina's Learned Sort becomes the fastest, followed by IS4o, with PCF Learned Sort being the slowest among the three. This outcome is not surprising, as IS4o is a highly optimized sample sort implementation and Ferragina's Learned Sort explicitly focus on cache-level efficiency, whereas our PCF Learned Sort is designed with an emphasis on providing rigorous theoretical guarantees rather than aggressive low-level optimizations.
> - **Robustness**: We observed that Ferragina's Learned Sort occasionally suffers from severe slowdowns under adversarial datasets. This is due to its fallback to insertion sort, which yields a worst-case complexity of $O(n^2)$. For example, for data of size $n=10^6$ drawn from a normal distribution with injected point masses (a setup described in a later section of the rebuttal), we obtained the following results (minimum and maximum sorting times over 10 runs):
>   - std::sort:  0.076 ~ 0.078 s
>   - PCF Learned Sort (Ours): 0.041 ~ 0.042  s
>   - Learned Sort 2.1 (Ferragina and Odorisio, 2025): 0.020 ~ **0.154** s
>
> These revisions are reflected in Section 4 (Experiments) and in Figures 2 and 4, where the new baselines have been incorporated.
>
> These experiments indicate that while PCF Learned Sort is often outperformed on average by highly optimized implementations, its theoretical worst-case guarantee ensures stable performance even under adversarial distributions. This robustness supports the main focus of our study, namely, providing a theoretical foundation for learned sorts. We believe such a foundation, when combined with optimized implementations, can pave the way for learned sorts that achieve both practicality and reliability.

---

> ### Author Response · Authors · 2025-09-09
> **Rebuttal by Authors**
>
> # Theoretical Assumptions and Robustness
>
> ## Distributional Assumptions
>
> > Weaknesses: The main theoretical result holds under a bounded probability density ($0<\sigma_1 <= f(x) <= \sigma_2 < \infty$), but there is limited discussion about the prevalence of such distributions in real-world data.
>
> > Questions: Could you clarify the typical prevalence of the required distributional assumption ($0<\sigma_1 <= f(x) <= \sigma_2 < \infty$) in practical, large-scale datasets? Are there natural, widely arising scenarios where this fails, and if so, could the algorithm or theory be adapted?
>
> We thank the reviewer for raising this important point.
> Our theoretical assumption ($0 < \sigma_1 \leq f(x) \leq \sigma_2 < \infty$) corresponds to the setting where the data lies within a finite domain and the probability density remains strictly positive and finite throughout that range. This assumption is quite reasonable and often holds in practice for large-scale datasets.
>
> For example, as discussed by Zeighami et al. (ICML 2023), this assumption naturally arises in domains where the range is inherently bounded, such as age, grades, or data over a period of time. Their experimental observations further suggest that the ratio $\sigma_2 / \sigma_1$ tends to remain close to 1 for many datasets (WL, IOT, BK, FB, WK), and even for more challenging datasets such as OSM, the ratio still appears to remain at most around 20.
>
> We acknowledge that there are cases where this assumption may fail, such as heavy-tailed distributions or data with extremely sparse regions. In such scenarios, our expected complexity guarantee does not hold; however, the algorithm still maintains the worst-case guarantee of $O(n \log n)$, ensuring robustness. This robustness is also supported by the additional experiments on adversarial datasets that we included in Appendix C (Experiments in Adversarial Environments) in the revised version.
>
> In the final version, we have expanded the discussion to cover the practical prevalence of this assumption, its exceptions, and the scope of our theoretical results (in Section 3.3). We are grateful to the reviewer for highlighting this crucial aspect.
>
>
> ## Adversarial and Mismatched Distributions
>
> > Weaknesses: There is no theoretical or experimental exploration of cases where the input distribution may be adversarial or poorly approximated by the CDF model, which would reveal algorithmic risk in edge cases.
>
> > Questions: In adversarial or model-mismatch settings (e.g., highly clustered or adversarially constructed data), does the bucketing lemma still hold, or do worst-case time complexities appear empirically even for non-pathological but practical datasets?
>
> We thank the reviewer for raising this important point.
> Indeed, for clustered datasets or adversarially constructed inputs, our lemma does not hold, and therefore our algorithm no longer achieves the expected complexity of $O(n \log \log n)$. However, the algorithm always retains the worst-case complexity guarantee of $O(n \log n)$. In other words, even under adversarial or mismatched distributions, the performance remains on par with traditional comparison-based sorting methods.
>
> To validate this, we conducted experiments using artificially clustered datasets that explicitly violate the assumption. Specifically, given an original dataset of length $n$, we injected $n$ duplicate values to create highly clustered data. The results were as follows:
> - For small $n$, the performance of PCF Learned Sort occasionally dropped to the level of std::sort.
> - In contrast, Learned Sort 2.0 (Kristo et al., 2021) sometimes exhibited severe slowdowns. For example, on data of size $n = 10^5$ sampled from a Normal distribution with injected duplicates, we observed the following (minimum and maximum sorting times over 10 runs):
>   - std::sort: 0.036 ~ 0.038 s
>   - PCF Learned Sort (Ours): 0.019 ~ 0.021 s
>   - Learned Sort 2.0 (Kristo et al. 2021): 0.014 ~ **108.9** s
>
> These results confirm that while the expected bound breaks under adversarial or mismatched inputs, the worst-case guarantee ensures stable performance without catastrophic slowdowns. This contrasts with methods lacking worst-case guarantees, such as Kristo et al. (2021), which may suffer orders-of-magnitude degradation. We believe this robustness represents a key strength of our approach. The results of these experiments are included in Appendix C of the final version.

---

> ### Author Response · Authors · 2025-09-09
> **Rebuttal by Authors**
>
> # Alternative CDF Models
>
> > Weaknesses: The paper settles on piecewise constant approximation, but does not deeply analyze or experiment with other popular CDF models (e.g., neural CDFs, histograms of differing bin widths, spline-based models).
>
> > Questions: Is there scope (and associated theory) for substituting the PCF with more expressive or data-adaptive CDF models (e.g., splines, neural networks) in this framework? Have you tried such models, and how do they fare empirically or theoretically?
>
> We thank the reviewer for raising this point. Our Learned Sort framework is not restricted to PCF; it is a general framework in which any CDF model (e.g., splines, neural networks) can be substituted.
>
> Following this suggestion, we additionally evaluated an implementation using spline-based models. Specifically, instead of using constant values per interval as in PCF, we constructed splines by linearly interpolating the values of the empirical CDF at the endpoints of each interval. The results were as follows:
> - On small-scale inputs or in adversarial settings, the spline-based model provided a more accurate approximation of the CDF, yielding more balanced buckets and outperforming PCF Learned Sort in some cases.
> - For large $n$, however, the increased cost of CDF prediction dominated, and performance degraded compared to PCF Learned Sort.
>
> We also proved that spline-based models defined in this way still satisfy the assumptions of Lemma 3.3, thereby preserving the expected complexity guarantee of $\mathcal{O}(n \log \log n)$. Concretely, we showed that with splines, $\Pr[\exists j, |c_j| > \lfloor n^d \rfloor] = \mathcal{O}(1 / \log n)$ still holds.
>
> While more complex models such as neural networks are in principle compatible with our framework, we did not adopt them. As noted in Kristo et al. (2020), training and inference overheads make heavy models impractical in the Learned Sort setting, and they also make it difficult to retain theoretical guarantees. Since the main focus of this work is on Learned Sort with provable expected and worst-case complexity, we did not pursue neural approaches.
>
> In the final version, we included the theoretical and empirical findings for spline-based models (in Section 3.3, Section 4, and Appendix A.5) as well as a discussion on the limitations of more complex alternatives (in Section 5).
>
>
> # Parallelism, Scalability, and Implementation Details
>
> > Weaknesses: there is no discussion (theoretical or experimental) of parallelization potential, memory overhead, or multi-core scaling, all of which have been central for modern sort implementations.
>
> > Questions: Are there any plans for public release of source code, and can you comment on practical implementation nuances such as memory usage or multi-core scaling?
>
> We appreciate the reviewer's comment and fully agree on the importance of parallelism.
> While the current paper is focused on theoretical analysis and thus does not include an experimental study of parallel implementations, the algorithm is structurally well-suited for parallelization. Specifically, training the CDF model, computing bucket IDs for each element, and distributing elements into buckets can all be performed in parallel. After bucketing is completed, the buckets are independent, and the final sorting step within each bucket can also proceed in parallel. This indicates that the design is inherently parallelizable, even though a practical evaluation of multi-core scalability is left for future work. We note that it is not uncommon for prior sorting studies to omit parallel implementations, so we do not believe this limitation diminishes the contributions of our work. To clarify this point, we have added an explicit discussion on the parallelizability of our algorithm in the revised version (in Section 5).
>
> Regarding code availability, our implementation will be publicly released to facilitate reproducibility and follow-up research.
>
> As for memory usage, a straightforward implementation requires allocating an additional buffer for buckets of roughly the same size as the input array. Furthermore, in our implementation, we store the bucket IDs (as 32-bit integers) to avoid recomputing predictions for the same element, improving performance. For example, sorting an array of $10^8$ doubles ($\approx$ 800 MB) requires: the input array ($\approx$ 800 MB), the bucket buffer ($\approx$ 800 MB), and the bucket ID storage ($\approx$ 400 MB), for a total memory footprint of about 2.0 GB.

---

> ### Author Response · Authors · 2025-09-09
> **Rebuttal by Authors**
>
> # Experimental Coverage
>
> ## Training Costs
>
> > Weaknesses: There is a brief acknowledgement that model training is factored into the complexity, but the impact of training time relative to inference/bucketing is not explored experimentally, which is practical for real-world high-throughput sorting.
>
> We thank the reviewer for highlighting this point.
> In response, we conducted additional experiments to measure the time contributions of each component of PCF Learned Sort (model training, bucketing, and the "standard" sort for too big/small buckets). The results revealed the following:
> - For small $n$, the majority of time is often spent in applying the "standard" sort to buckets smaller than $\tau$. For large $n$, most of the time is instead consumed by bucketing.
> - Training time is always comparable to or shorter than the bucketing time, never exceeding about one quarter of the total runtime, and thus does not become the dominant factor.
> - In special cases (e.g., adversarial inputs or datasets with very high duplication), a larger fraction of time is spent in sorting large buckets, as bucketing is more likely to "fail" in such scenarios.
>
> These analyses clarify which parts of our algorithm become the bottleneck under different conditions. We believe this provides important practical insight into the method. In the final version, we added the results as Appendix D and briefly summarized them in the main text (in Section 4).
>
>
> ## Dataset Diversity
>
> > While several real datasets are used, broader inclusion of highly skewed, real-world distributions (not just OSM/Books, which are discussed) would help assess generality.
>
> We thank the reviewer for this helpful suggestion.
> Following prior work, we expanded our evaluation to include 12 additional real-world datasets, bringing the total to 16 datasets. The newly added datasets are: Face, NYC [Dist, Tot], SOF [Humidity, Pressure, Temp], Chicago [Start, Tot], and Stocks [Volume, Open, Date, Low]. These cover a wide variety of distributions, many of which are highly skewed (see the histograms in Figure 2 of the revised manuscript).
>
> Across this broader set of datasets, we consistently observed that PCF Learned Sort achieves sub–$n \log n$ expected complexity. On heavily duplicated data (such as NYC [Dist, Tot], SOF [Humidity, Temp], Chicago [Tot], and Stocks [Volume, Open, Date, Low]), buckets often reduce to a single value very early, so PCF Learned Sort tends to finish quickly.
>
> More importantly, these experiments revealed vulnerabilities in existing methods. For example, on the SOF [Temp] dataset with medium input sizes ($2 \times 10^5 < n \leq 2 \times 10^7$), we observed dramatic slowdowns in Learned Sort 2.0 (Kristo et al., 2021). For example, when $n = 2 \times 10^7$, PCF Learned Sort consistently completed in about 0.45 seconds, whereas Learned Sort 2.0 (Kristo et al., 2021) sometimes required up to 326.4 seconds.
>
> These revisions are reflected in Section 4 (Experiments) and in Figures 2 and 4, where the new datasets have been incorporated.
>
> These findings highlight that existing Learned Sort approaches, which lack worst-case guarantees, can suffer severe slowdowns on real-world data. By contrast, our method retains both expected and worst-case complexity guarantees, demonstrating robust and consistent performance across diverse datasets.

---

### Decision · Action_Editor_7ako · 2025-10-22

**Recommendation:** Accept as is

**Audience:**

Yes

**Audience Explanation:**

Given that sorting is a fundamental operation and the power of learnable sort, this paper should be of broad interest: both to algorithmic and data structure community.


This is clearly one of the strongest paper that I have handled as an editor.

**Claims And Evidence:**

Yes

**Claims Explanation:**

Sorting is a fundamental operation in software engineering and for which the optimal algorithm has been known for a long time; the optimality here is for the worst case of input. This paper focuses on learnable sort which can utilize the underlying data distribution; under fairly mild but possibly general enough assumption, they show that the complexity can be reduced to n log log n (which is linear for all practical purposes!). The authors also provide empirical evidence to support their claim.

---

> ### Author Response · Authors · 2025-10-24
> **Camera-Ready Version Submitted**
>
> Dear Prof. Meel,
>
> We are deeply grateful for your decision to accept our paper and to award it the Featured Certification. We would also like to thank the reviewers for their constructive feedback, which has greatly helped us improve our work.
>
> We have now submitted the camera-ready version.
>
> Best regards,
> The authors of Paper 4387